# Neonatal gene therapy achieves sustained disease rescue of maple syrup urine disease in mice

Clément Pontoizeau[1,2,3✉], Marcelo Simon-Sola[3], Clovis Gaborit[3], Vincent Nguyen [3], Irina Rotaru[3], Nolan Tual[3], Pasqualina Colella [4], Muriel Girard[5,6], Maria-Grazia Biferi[7], Jean-Baptiste Arnoux[2], Agnès Rötig [3], Chris Ottolenghi[1,2,3], Pascale de Lonlay[2,6], Federico Mingozzi[4], Marina Cavazzana [3,8] & Manuel Schiff [2,3✉]

Maple syrup urine disease (MSUD) is a rare recessively inherited metabolic disorder causing accumulation of branched chain amino acids leading to neonatal death, if untreated. Treatment for MSUD represents an unmet need because the current treatment with life-long low-protein diet is challenging to maintain, and despite treatment the risk of acute decompensations and neuropsychiatric symptoms remains. Here, based on significant liver contribution to the catabolism of the branched chain amino acid leucine, we develop a liver-directed adeno-associated virus (AAV8) gene therapy for MSUD. We establish and characterize the *Bckdha* (*branched chain keto acid dehydrogenase a*)$^{-/-}$ mouse that exhibits a lethal neonatal phenotype mimicking human MSUD. Animals were treated at P0 with intravenous human *BCKDHA* AAV8 vectors under the control of either a ubiquitous or a liver-specific promoter. *BCKDHA* gene transfer rescued the lethal phenotype. While the use of a ubiquitous promoter fully and sustainably rescued the disease (long-term survival, normal phenotype and correction of biochemical abnormalities), liver-specific expression of *BCKDHA* led to partial, though sustained rescue. Here we show efficacy of gene therapy for MSUD demonstrating its potential for clinical translation.

[1] Necker Hospital, APHP, Biochemistry, Metabolomics Unit, Paris Cité University, Paris, France. [2] Necker Hospital, APHP, Reference Center for Inborn Error of Metabolism, Pediatrics Department, Paris Cité University, Filière G2M, Paris, France. [3] Inserm UMR_S1163, Institut Imagine, Paris, France. [4] Généthon INTEGRARE UMR-S951, University of Evry, Evry, France. [5] Necker Hospital, APHP, Pediatric Hepatology Unit, Pediatrics Department, Paris Cité University, Paris, France. [6] Inserm U1151, Institut Necker Enfants Malades, Paris, France. [7] Sorbonne University, Inserm, Institute of Myology, Centre of Research in Myology, Paris, France. [8] Necker Hospital, APHP, Biotherapies Department, Paris Cité University, Paris, France. ✉email: clement.pontoizeau@aphp.fr; manuel.schiff@aphp.fr

Monogenic inborn errors of metabolism are relevant targets for gene therapy as they are autosomal recessive conditions with well-characterized pathophysiology and highly informative biochemical readouts; yet their therapeutic management remains complex and cumbersome. Maple syrup urine disease (MSUD, MIM: 248600) is one of the earliest described metabolic disorders. This autosomal recessive disorder with an incidence of one in 185,000 live births is caused by a defective activity of the branched-chain 2-keto acid dehydrogenase (BCKD) enzyme leading to the accumulation of branched-chain amino acids (BCAA) leucine, isoleucine, valine and their corresponding 2-ketoacids (BCKA) in tissues and body fluids[1,2]. The BCKD enzyme is a multimeric enzyme complex with four components, branched-chain keto acid decarboxylase alpha and beta subunits (E1$_\alpha$ and E1$_\beta$), dihydrolipoyl transacylase (E2) subunit, and dihydrolipoamide dehydrogenase (E3) subunit. MSUD is caused by mutations in BCKDHA, BCKDHB or DBT genes respectively coding for E1$_\alpha$, E1$_\beta$, and E2 subunits and accounting for 45, 35, and 20% of MSUD patients, respectively[2]. Neurotoxicity in MSUD is related to the accumulation of leucine and 2-ketoisocaproic acid (KIC, the ketoacid derived from leucine)[3]. In the classical severe form of MSUD (85–95% of cases[4]), with less than 3% residual enzyme activity, this accumulation causes coma and cerebral edema shortly after birth with early death in the absence of aggressive and rapid management.

Long-term MSUD management represents an unmet clinical need[4]. Current MSUD treatment is limited to severe and life-long BCAA dietary restriction associated with an oral BCAA-free amino acids mixture. Such treatment is difficult to maintain in the long term and is largely incompatible with a normal quality of life. Further, it does not prevent long-term neurocognitive[5] and psychiatric impairments[6], the latter affecting roughly 30% of the patients after the age of 20 years[6]. Orthotopic liver transplantation (OLT), was shown as an effective MSUD therapy allowing removal of dietary restrictions, full protection from acute decompensations during illness[7,8], arrest of neurocognitive impairment progression[3,9], prevention of life-threatening cerebral edema[3], metabolic and clinical stability[9]. However, liver transplantation is associated with the potential risk of death and graft failures[9]. As an autosomal recessive monogenic disease, MSUD represents an ideal target for liver-directed gene therapy since clinical OLT data suggests that incomplete restoration of liver BCKD enzyme activity (representing 9–13% of body BCKD activity[10]) is fully therapeutic. This constitutes a strong rationale for testing liver gene transfer as a therapeutic option for MSUD.

Among the available gene-delivery platforms, adeno-associated virus (AAV) vectors are the most suitable for liver gene transfer[11]. AAV liver gene therapy achieved a major milestone with the proof of safety and long-term efficacy in a clinical trial for hemophilia B[12]. Inborn errors of metabolism are good candidates for AAV gene therapy[13]. Proof of concept of efficacy was obtained in mice for urea cycle disorders[14–17], organic acidemias[18], phenylketonuria[19], and others, and clinical trials of liver gene transfer are currently being conducted for a number of diseases, including ornithine transcarbamylase deficiency (OTC) (NCT02991144), glycogen storage disease type 1a (NCT03517085), mucopolysaccharidosis type VI (MPSVI) (NCT03173521), and Pompe disease (NCT03533673 and NCT04093349).

Mouse models of classical and intermediate MSUD with mutations in Dbt have been developed and characterized[20–22]. While the majority of patients harbor mutations in BCKDHA and BCKDHB genes, there is no characterized mouse model of MSUD linked to the Bckdhb gene, and only one report of a Bckdha knockout mouse with a severe phenotype and lethality within 24 h after birth and elevated BCAA and BCKA in plasma and tissues[23]. Recently, a mouse model with tissue-specific Bckdha knockout in brown adipose tissue was described and showed a reduced tolerance of BCAA loading but no other phenotypic features of MSUD[24].

We herein extensively characterized a Bckdha$^{-/-}$ mouse line showing that it faithfully recapitulates the classical human MSUD. We further developed a successful AAV gene therapy based on the transfer of human BCKDHA (hBCKDHA) in neonatal Bckdha$^{-/-}$ mice.

## Results

**Bckdha$^{-/-}$ mouse recapitulates the severe human phenotype of MSUD.** We generated Bckdha$^{-/-}$ mice by crossing publicly available heterozygous Bckdha$^{+/-}$ males and females, who did not display any particular phenotype. We found that Bckdha$^{-/-}$ mice show a lethal early-onset phenotype. Half of the mice died before 3 days of age, and the other half around 7 days of age with a maximum lifespan of 12 days (Fig. 1a). Bckdha$^{-/-}$ mice exhibited a major growth delay (Fig. 1b). Animals who survived longer than 1 week showed reduced activity and abnormal hindlimb clasping in response to tail suspension (100% of these short-survivors exhibited a score of 0 compatible with severe neurological impairment[25]). Biochemical analyses showed increased concentrations of plasma BCAA (Fig. 1c and Fig. S1a), along with a significant accumulation of alloisoleucine, a pathognomonic marker of MSUD (Fig. 1c), and increased concentrations of plasma BCKA (Fig. S1b). Conversely, alanine and glutamate plasma concentrations were decreased, leading to an increase in the leucine/alanine ratio (Fig. S1a) as described in MSUD patients[4]. RT-PCR showed a decrease of Bckdha transcripts in the liver, heart, and brain to about 50% in Bckdha$^{+/-}$ mice relative to Bckdha$^{+/+}$, whereas Bckdha transcripts were barely or not detectable in Bckdha$^{-/-}$ mice in all of these tissues (Fig. 1d). Western blot showed similar results for BCKDHA protein in the liver, heart, and brain, confirming the null nature of the model (Fig. 1e). Furthermore, BCKDHB protein was undetectable in the liver of Bckdha$^{-/-}$ mice as described in BCKDHA defective lymphoblasts from MSUD patients[26] (Fig. S1c). Together, these results indicate the Bckdha$^{-/-}$ mouse model faithfully recapitulates the features of human classical MSUD.

**Design and in vitro testing of cassettes for the treatment of MSUD.** Transfection of plasmids in Huh7 cells resulted in the expression of BCKDHA for all transgenes with the highest expression detected with the wild-type sequence as assessed by Western blot analysis (Fig. 2b, d). We therefore selected the EF1α (ubiquitous promoter) and hAAT (human alpha 1 antitrypsin, liver-specific promoter) hBCKDHA transgenes with the full-length human BCKDHA cDNA sequence (Fig. 2a, c). Accordingly, codon-optimized constructs were not further used in the study. To ensure that the difference of intron between the EF1α driven construct (containing the EF1α intron) and the hAAT driven construct (containing the SV40 intron) did not affect BCKDHA expression, we designed an additional hAAT driven cassette containing the EF1α intron. Similar levels of BCKDHA protein expression in transfected Huh7 cells between the two constructs (hAAT-iEF1A and hAAT iSV40) were obtained (Fig. S2a). As it was initially reported that 80% of BCKD activity was provided by the liver in rodents[10], we opted for AAV8 vectors given their ability to mainly transduce the liver in mice, along with other tissues such as the heart[27].

**AAV8-EF1α-hBCKDHA gene transfer allows for rescue of MSUD early lethality in Bckdha$^{-/-}$ mice.** To determine the minimally effective vector dose that would rescue the MSUD phenotype in Bckdha$^{-/-}$ mice, we performed systemic temporal

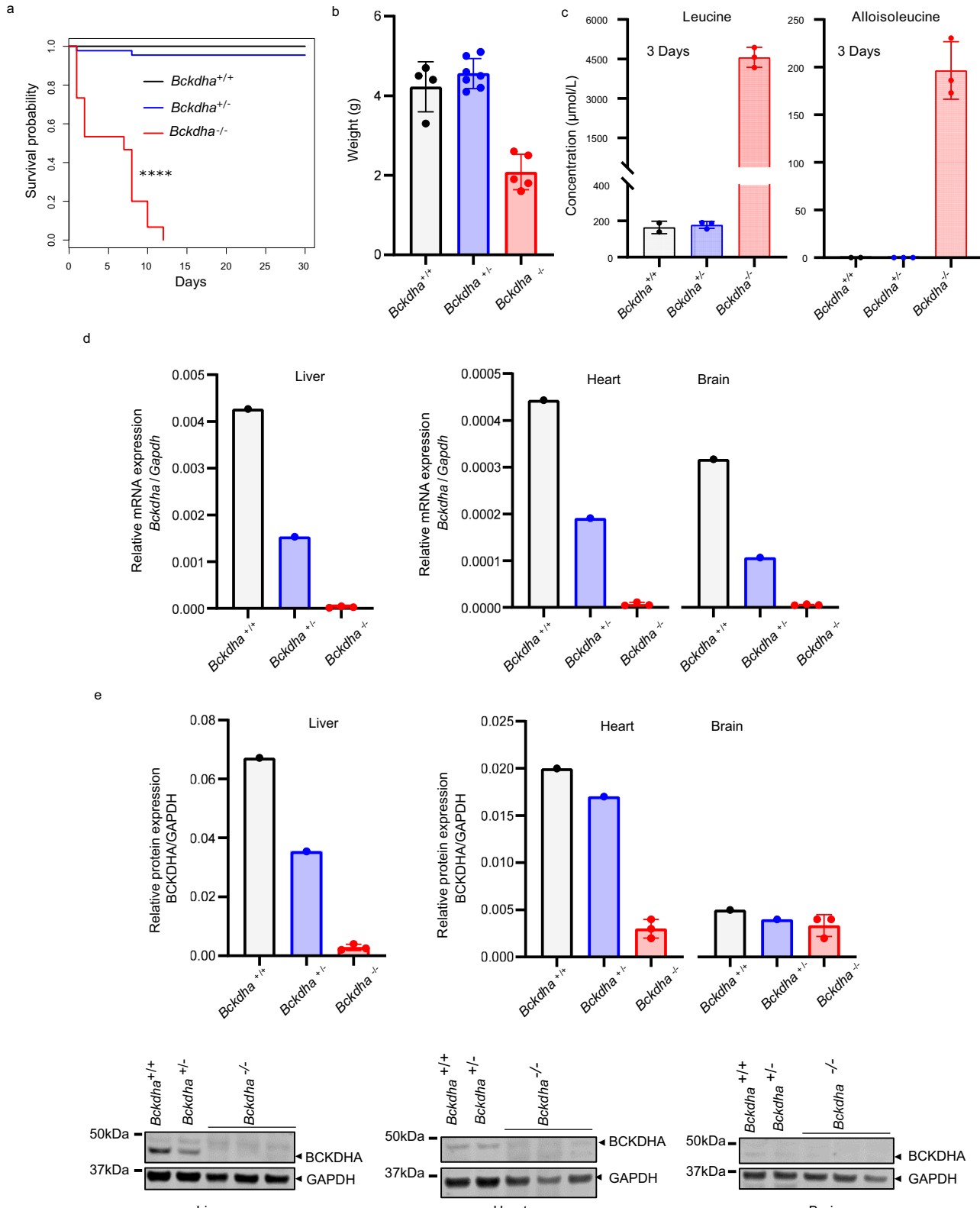

**Fig. 1 The *Bckdha*$^{-/-}$ mouse recapitulates the MSUD phenotype in humans. a** Kaplan–Meier curve (*Bckdha*$^{-/-}$ $n = 15$, *Bckdha*$^{+/-}$ $n = 44$, *Bckdha*$^{+/+}$ $n = 26$); log-rank Mantel–Cox test (*Bckdha*$^{-/-}$ vs. *Bckdha*$^{+/+}$ or *Bckdha*$^{+/-}$, $p$ values = $4.86 \times 10^{-13}$ or $2.22 \times 10^{-16}$, respectively). **b** Weight (1 week, *Bckdha*$^{-/-}$ $n = 5$, *Bckdha*$^{+/-}$ $n = 7$, *Bckdha*$^{+/+}$ $n = 4$). **c** Leucine and alloisoleucine plasma concentrations (3 days, *Bckdha*$^{-/-}$ $n = 3$, *Bckdha*$^{+/-}$ $n = 3$, *Bckdha*$^{+/+}$ $n = 2$). **d** *Bckdha* mRNA and **e** BCKDHA protein levels in the liver, heart, and brain (1 week, *Bckdha*$^{-/-}$ $n = 3$, *Bckdha*$^{+/-}$ $n = 1$, *Bckdha*$^{+/+}$ $n = 1$). All data are shown as means ± SD. ****$P < 0.0001$. Source data are provided as a Source data file.

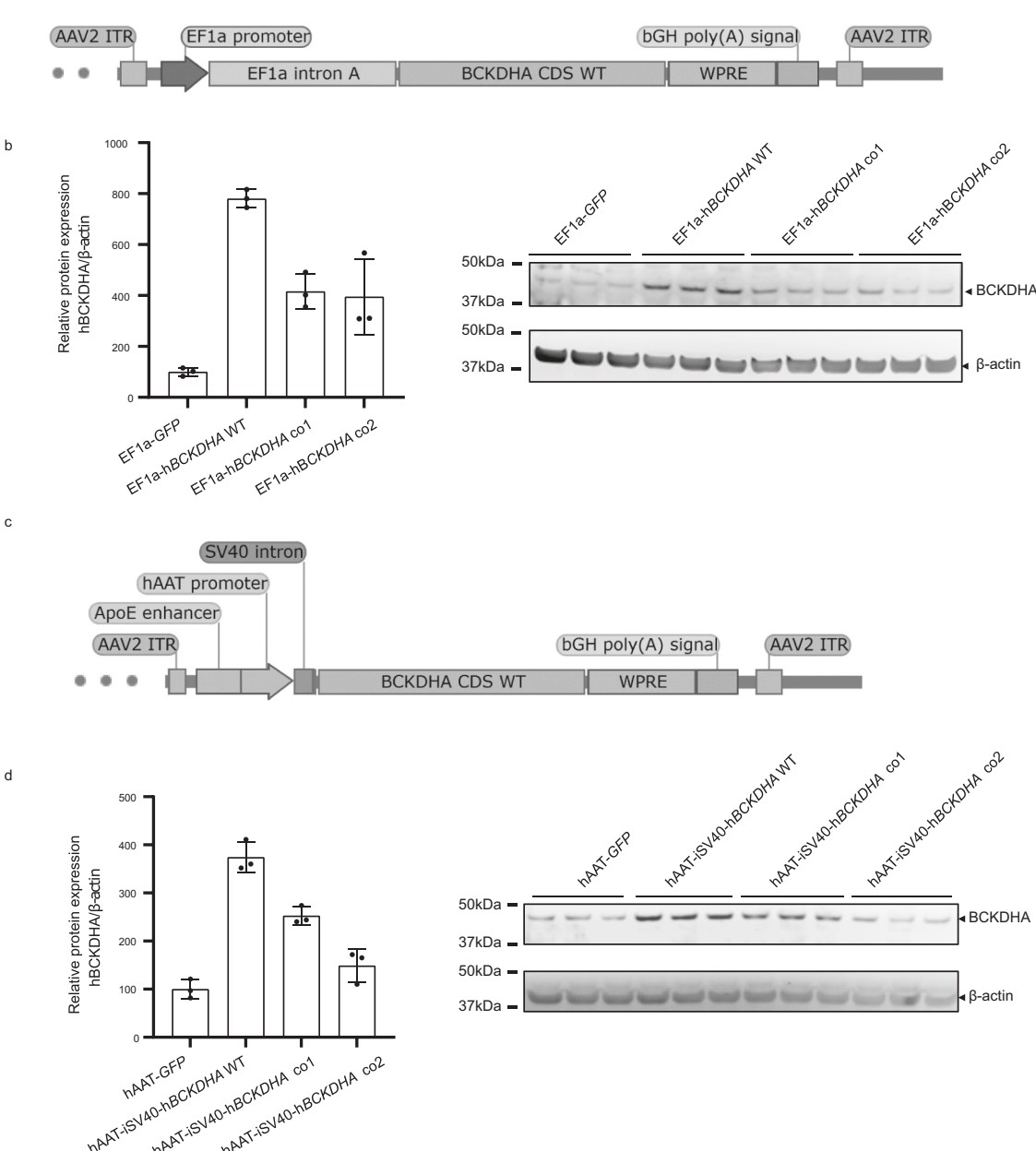

**Fig. 2 Schematic representation of vector expression cassettes used in the study. a** AAV8-EF1α-h*BCKDHA* vector genome consisting of an expression cassette including a ubiquitous human elongation factor 1-alpha promoter (hEF1a) with an extra intronic sequence, the h*BCKDHA* coding sequence, a WPRE sequence, and a polyadenylation signal (bGH poly(A)), flanked by AAV2 ITRs. **b** Western blot analyses of human BCKDHA in Huh7 cells transfected with EF1α *GFP*, EF1α h*BCKDHA* WT, EF1α h*BCKDHA* co1 (codon-optimized sequence 1) and EF1α h*BCKDHA* co2 (codon-optimized sequence 2) ($n = 3$ biological replicates for each condition). **c** AAV8-hAAT-h*BCKDHA* vector genome consisting of an expression cassette including hepatocyte-specific hAAT promoter composed of the ApoE enhancer and the human *alpha* 1 *antitrypsin promoter* (hAAT), a SV40 intronic sequence, the h*BCKDHA* coding sequence, a WPRE sequence and a polyadenylation signal (bGH poly(A)), flanked by AAV2 ITRs. **d** Western blot analyses of human BCKDHA in Huh7 cells transfected with hAAT *GFP*, hAAT iSV40 h*BCKDHA* WT, hAAT iSV40 h*BCKDHA* co1 (codon-optimized sequence 1), and hAAT iSV40 h*BCKDHA* co2 (codon-optimized sequence 2), the three of them carrying the SV40 intron ($n = 3$ biological replicates for each condition). All data are shown as means ± SD. Source data are provided as a Source data file.

vein injection of the AAV8-EF1α-h*BCKDHA* vector at either $10^{13}$ vg/kg ($1.5 \times 10^{10}$ vg/mouse, further referred to as low dose) or $10^{14}$ vg/kg ($1.5 \times 10^{11}$ vg/mouse further referred to as high dose) at P0, immediately after birth. All the pups of the litters were injected, without prior knowledge of the genotyping data, which were obtained at P10. Three litters were injected at P0 with $10^{13}$ vg/kg and two litters with $10^{14}$ vg/kg of the AAV8-EF1α-h*BCKDHA* vector. In the litters injected at $10^{13}$ vg/kg, one pup died at P3 (eaten by the dam), one *Bckdha*$^{-/-}$ pup died at P1, and

one at P7 (traumatic urine collection), leaving for the analysis 7 *Bckdha*$^{-/-}$, 9 *Bckdha*$^{+/-}$ and 5 *Bckdha*$^{+/+}$ mice. In the litters injected at $10^{14}$ vg/kg, two *Bckdha*$^{-/-}$ died at P2 and one heterozygous *Bckdha*$^{+/-}$ died at P7 in a context of major growth failure, leaving for the analysis 2 *Bckdha*$^{-/-}$, 9 *Bckdha*$^{+/-}$ and 4 *Bckdha*$^{+/+}$ mice. With the injections at $10^{13}$ vg/kg, we observed a partial and transient rescue of the MSUD phenotype in *Bckdha*$^{-/-}$ mice ($n = 7$) with important inter-individual variability. Five out of 7 treated *Bckdha*$^{-/-}$ mice showed normal

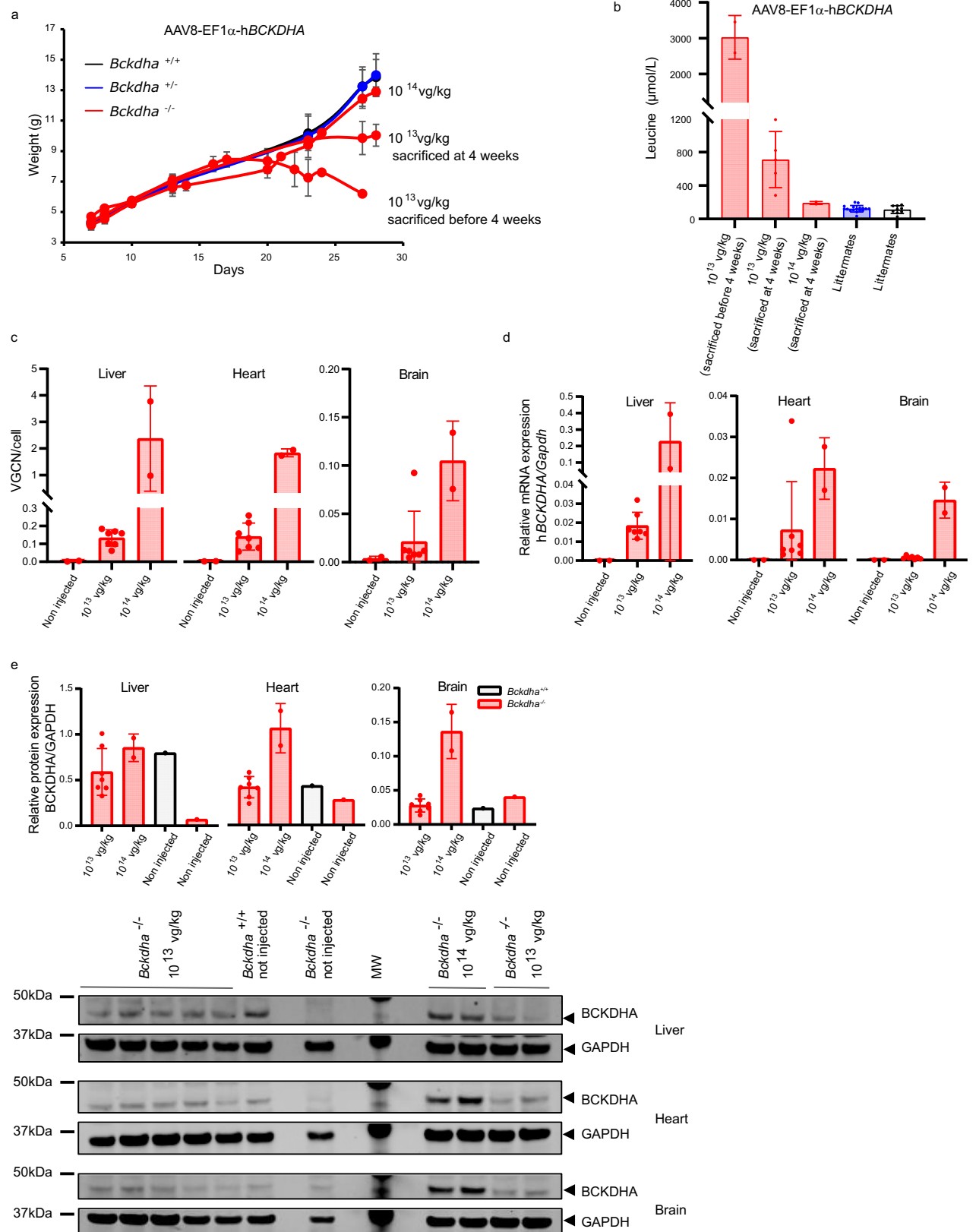

growth without obvious neurological symptoms during the first 3 weeks post vector infusion, but then stopped growing and developed neurological symptoms (ataxia with frequent falls), urging to sacrifice them at age 4 weeks (Fig. 3a). In the remaining 2/7 $Bckdha^{-/-}$ mice growth retardation was obvious very early,

evolving into weight loss and finally a moribund status requiring an anticipated sacrifice before age 4 weeks (Fig. 3a). The $Bckdha^{-/-}$ mice injected with the AAV8-EF1α-h$BCKDHA$ vector at $10^{14}$ vg/kg displayed normal growth without overt neurological symptoms (Fig. 3a $n = 2$ $Bckdha^{-/-}$ injected at $10^{14}$ vg/kg and

**Fig. 3 AAV8-EF1α-h*BCKDHA* gene transfer allows for rescue of MSUD early lethality in *Bckdha*−/− mice. a** Weight curves and **b** plasma leucine concentrations at sacrifice (*Bckdha*−/− injected with $10^{13}$ vg/kg and sacrificed <4 weeks $n = 2$ (2 females), *Bckdha*−/− injected with $10^{13}$ vg/kg and sacrificed at 4 weeks $n = 5$ (1 male/4 females), *Bckdha*−/− injected with $10^{14}$ vg/kg and sacrificed at 4 weeks $n = 2$ (1 male/1 female), *Bckdha*+/− $n = 18$ (9 males/9 females), *Bckdha*+/+ $n = 9$ (2 males/7 females). The blood sample of a *Bckdha*−/− mice was insufficient for analysis. **c** Vector genome copy number (VGCN), **d** human *BCKDHA* (h*BCKDHA*) mRNA(*Bckdha*−/− injected at $10^{13}$ vg/kg $n = 7$, *Bckdha*−/− injected at $10^{14}$ vg/kg $n = 2$, controls: *Bckdha*−/− non injected and sacrificed at 1 week $n = 2$), and **e** BCKDHA protein levels in the liver, heart, and brain in mice sacrificed around 4 weeks after neonatal injection (*Bckdha*−/− injected at $10^{13}$ vg/kg $n = 7$, *Bckdha*−/− injected at $10^{14}$ vg/kg $n = 2$, controls: *Bckdha*−/− $n = 1$ non-injected and sacrificed at 1 week + 1 non-injected *Bckdha*+/+ sacrificed at 4 weeks); analyses were performed in triplicate. The antibody detected both the human BCKDHA and the murine BCKDHA proteins. All data are shown as mean ± SD. Source data are provided as a Source data file.

sacrificed at age 4 weeks). We observed a similar dose effect at the biochemical level, with the *Bckdha*−/− mice injected at $10^{14}$ vg/kg displaying plasma leucine concentrations slightly higher than the upper limit of the normal range, whereas the *Bckdha*−/− mice injected at $10^{13}$ vg/kg and sacrificed at age 4 weeks displayed a marked six-fold increase the normal in leucine concentrations (Fig. 3b). The two phenotypically most severe *Bckdha*−/− mice injected at $10^{13}$ vg/kg and sacrificed before age 4 weeks exhibited extremely high plasma leucine concentrations, consistent with their clinical status, suggesting a loss of treatment efficacy and a relapse of the disease. AAV8 vector genome copies were detectable in the liver and heart and more moderately in the brain with a significant dose effect between injections at $10^{13}$ and $10^{14}$ vg/kg (Fig. 3c). In the liver and heart, h*BCKDHA* mRNA transcripts and hBCKDHA protein were detectable with a similar dose effect (Fig. 3d, e). In brain, h*BCKDHA* mRNA transcripts and hBCKDHA protein were only detectable for injections at $10^{14}$ vg/kg (Fig. 3d, e). Liver pathology of injected mice did not show significant alteration (Fig. S3). In the brain of *Bckdha*−/− mice sacrificed before age 4 weeks due to relapse of the disease, we observed an important vacuolization of the striatum (Fig. S3).

These results indicate that rescue or MSUD in mice can be achieved with systemic gene transfer with AAV vectors.

**High dose AAV8-EF1α-h*BCKDHA* gene transfer allows long-term sustained MSUD rescue.** To further evaluate the long-term effect of gene therapy in MSUD mice, we next injected the AAV8-EF1α-h*BCKDHA* vector in nine *Bckdha*−/− pups from 3 litters at $10^{14}$ vg/kg. Compared to their wild-type and heterozygous littermates, treated animals exhibited comparable survival and normal growth (Fig. 4a, b). All nine animals were alive at 6 months of age without overt phenotypic abnormalities. The biochemical phenotype was dramatically improved with BCAA concentrations that were reduced from twenty-fold to only two-fold over normal at age 6 months (Figs. 4c and S4) along with a reduction of BCKA concentrations (Fig. S5a). Alloisoleucine was not detectable. Other disturbances of amino acids such as decreased alanine and glutamate concentrations were near completely corrected (Fig. S4). We sacrificed 5 treated *Bckdha*−/− mice at 6 months of age. In these mice, AAV8 vector genome copies were detectable in the liver and heart and, at lower levels, in the brain (Fig. 4d). h*BCKDHA* mRNA transcripts were detectable in the liver and heart and, to a lesser extent, in the brain (Fig. 4e). Consistently, hBCKDHA protein was detectable mainly in the liver and heart and present albeit at lower levels in the brain (Fig. 4f). In the liver, hBCKDHA protein was less phosphorylated in *Bckdha*−/− treated mice than in *Bckdha*+/+ mice, suggestive of an important active BCKDHA fraction (Fig S5b). BCKDHB protein expression was also rescued in the liver to about 15% of its expression in the liver of wild-type individuals (Fig. S5c). Brain pathology did not show significant alterations as opposed to striatum vacuolization observed in *Bckdha*−/− mice treated with $10^{13}$ vg/kg and sacrificed at age 4 weeks (Fig. S3). The remaining four *Bckdha*−/− mice were still

alive without overt phenotypic abnormalities (Fig. S6a, b) at age 12 months with stable, near-normal leucine concentrations in plasma (Fig. S6c). In these 12 months old mice, AAV8 vector genome copies and hBCKDHA protein were detectable in the liver and heart and, at lower levels, in the brain (Fig. S6d, e). Other features such as the phosphorylation state of BCKDHA, partial rescue of BCKDHB protein expression, and correction of other amino acid disturbances were similar between 6- and 12-month-old (treated) *Bckdha*−/− individuals (Figs. S6f, g and S4). These results indicate that constitutive expression of the h*BCKDHA* transgene mediates long-term rescue of MSUD in mice.

**Liver-restricted gene transfer allows partial but sustained rescue of the MSUD phenotype.** We next wanted to evaluate the liver vs. extrahepatic contribution to the total BCKDHA activity responsible for the phenotypic rescue of mice treated with the AAV8-EF1α-h*BCKDHA* vector. To this aim, we tested the therapeutic efficacy of a vector expressing the h*BCKDHA* transgene under the control of a liver-specific promoter (AAV8-hAAT-h*BCKDHA*). Compared with the AAV8-EF1α-h*BCKDHA* low dose (see above), the $10^{13}$ vg/kg dose of AAV8-hAAT-h*BCKDHA* led to the transient rescue of the MSUD phenotype, as the 5/5 *Bckdha*−/− mice survived for 14 days, either without overt clinical symptoms (3/5) or with delayed though progressive growth (2/5). Thereafter, all mice became rapidly moribund thus requiring sacrifice between P19 and 21 (Fig. S7a). At sacrifice, mice showed a significant increase in plasma leucine, consistent with the severe phenotype, suggesting reduced treatment efficacy (Fig. S7b). We observed a comparable biodistribution of AAV8 in tissues similar to the previous experiments, with the detection of vector genome copies in the liver and heart and, at lower levels, in the brain (Fig. S7c). Results of mRNA and protein analyses were consistent with the hAAT promoter liver specificity (Fig. S7d, e). Therefore, the effects of low dose AAV8-hAAT-h*BCKDHA* were partly effective yet significantly less than low dose AAV8-EF1α-h*BCKDHA*.

Subsequently, we tested the intermediate ($3 × 10^{13}$ vg/kg) and high ($10^{14}$ vg/kg) doses of AAV8-hAAT-h*BCKDHA* along with parallel experiments testing an intermediate dose of AAV8-EF1α-h*BCKDHA*. The 5 *Bckdha*−/− mice that survived beyond 48 hours (out of a total of 9 pups) injected with $3 × 10^{13}$ vg/kg AAV8-hAAT-h*BCKDHA* vector had 3 weeks of normal growth without overt phenotypic abnormalities. Then they all exhibited abrupt growth arrest, severe neurological impairment with frequent falls, ataxic gait, all with a score of 0 at the hindlimb clasping test (Fig. S8a). This led to sacrifice during the fourth week, 1 week later than the animals treated at $10^{13}$ vg/kg (Fig. S7a). Conversely, only 1 of the 6 *Bckdha*−/− mice injected with $3 × 10^{13}$ vg/kg AAV8-EF1α-h*BCKDHA* vector that survived beyond 48 h (out of a total of 8 pups, plus 2 dead pups that could not be genotyped) died at 5 weeks post treatment displaying poor growth and ataxic gait. Three of the 5 other pups died or required sacrifice at 9, 10, or 12 weeks of age (Fig. S8c, d), whereas the two remaining

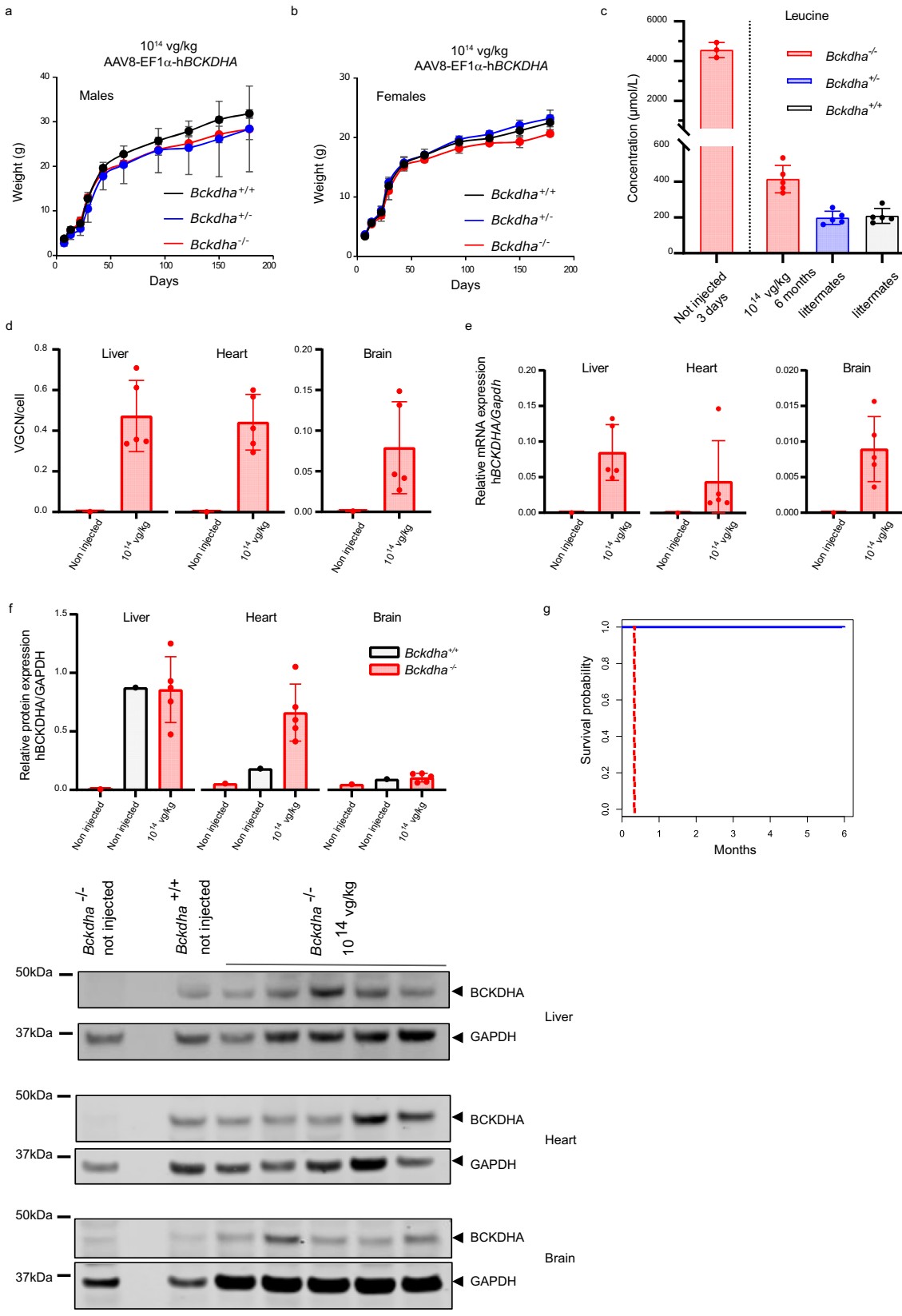

animals were in good health when they were sacrificed at age 12 weeks at the end of the study. Leucine plasma concentration measured at age 4 weeks in AAV8-hAAT-h*BCKDHA* treated animals (intermediate dose) was $3460 +/- 750\,\mu\text{mol/L}$ vs. $890 +/- 340\,\mu\text{mol/L}$ for *Bckdha*$^{-/-}$ mice in the AAV8-EF1α-h*BCKDHA* intermediate-dose cohort (Fig. S8b, e). These differences agreed with the less severe phenotype observed in AAV8-EF1α-h*BCKDHA*-treated mice. At age 8 weeks, the mean leucine plasma concentration for the *Bckdha*$^{-/-}$ mice in the intermediate dose EF1α group was $1070 +/- 230\,\mu\text{mol/L}$ indicating sustained moderate restoration of BCKDHA activity. Tissue analyses confirmed the expression and distribution of BCKDHA

**Fig. 4 Long-term sustained complete rescue of severe MSUD phenotype in *Bckdha*⁻/⁻ mice.** Mice were injected at $10^{14}$ vg/kg with AAV8-EF1α-h*BCKDHA*. **a** Weight curves for males (*Bckdha*⁻/⁻ $n = 6$, *Bckdha*⁺/⁻ $n = 3$, *Bckdha*⁺/⁺ $n = 3$) and **b** females (*Bckdha*⁻/⁻ $n = 3$, *Bckdha*⁺/⁻ $n = 8$, *Bckdha*⁺/⁺ $n = 6$) for all injected mice. **c** Plasma leucine concentrations for non-injected *Bckdha*⁻/⁻ newborns ($n = 3$) and at 6 months for injected mice, *Bckdha*⁻/⁻, *Bckdha*⁺/⁻, and *Bckdha*⁺/⁺ ($n = 5$ each). **d** Vector genome copy number (VGCN), **e** human *BCKDHA* (h*BCKDHA*) mRNA, and **f** BCKDHA protein levels in the liver, heart, and brain in mice sacrificed 6 months after neonatal injection (*Bckdha*⁻/⁻ injected $n = 5$, controls: *Bckdha*⁻/⁻ $n = 1$ non-injected and sacrificed at 1 week + 1 non-injected *Bckdha*⁺/⁺ sacrificed at 6 months for western blots). Analyses were performed in triplicate. The antibody detected both the human BCKDHA and the murine BCKDHA proteins. **g** Kaplan–Meier curves for all injected *Bckdha*⁻/⁻ mice ($n = 9$) (blue line) during the first 6 months. The red dash line represents the maximum survival of untreated *Bckdha*⁻/⁻ mice. All data are shown as mean ± SD. Source data are provided as a Source data file.

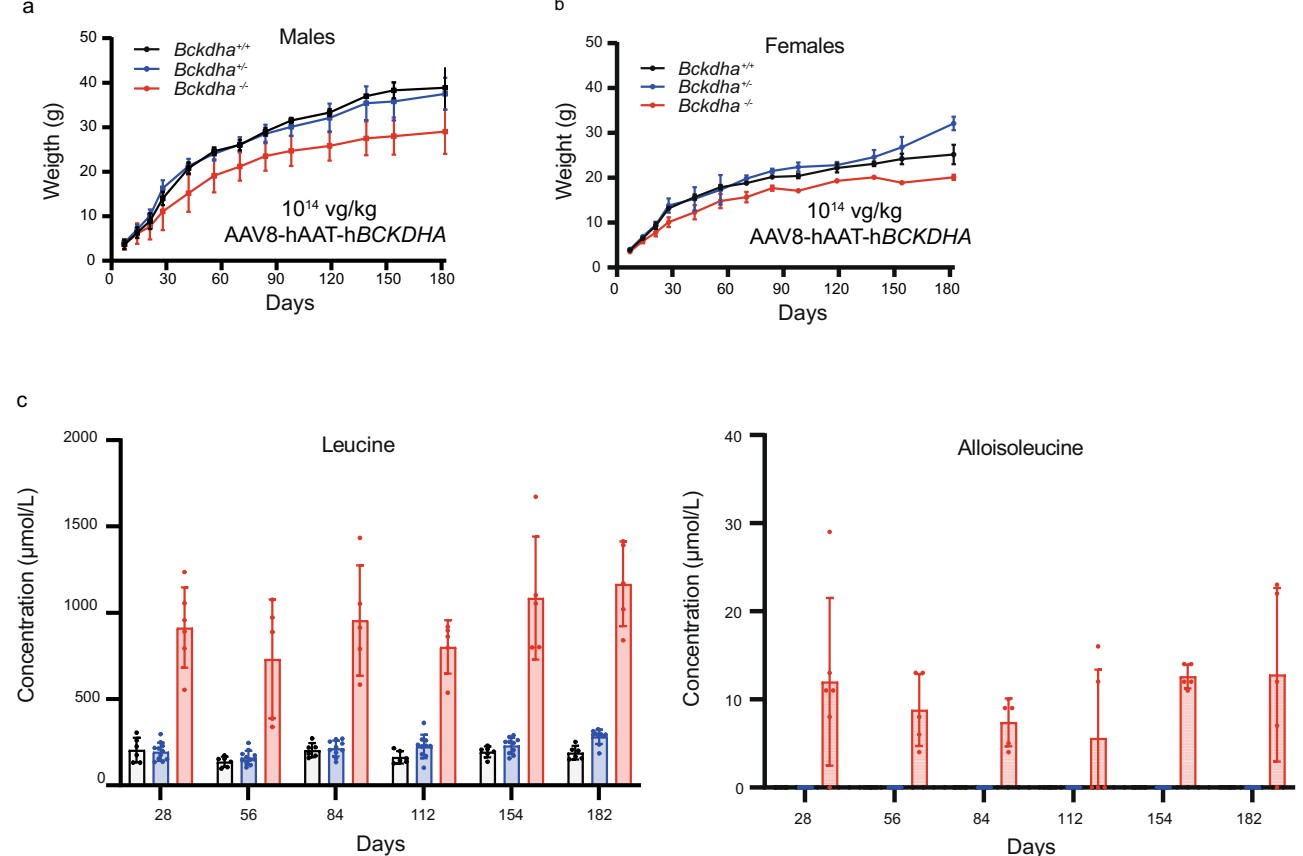

**Fig. 5 Long-term partial rescue of severe MSUD phenotype in *Bckdha*⁻/⁻ mice injected with AAV8-hAAT-h*BCKDHA* at $10^{14}$ vg/kg. a** Weight curves for males (*Bckdha*⁻/⁻ $n = 4$, *Bckdha*⁺/⁻ $n = 6$, *Bckdha*⁺/⁺ $n = 2$) and **b** females (*Bckdha*⁻/⁻ $n = 2$, *Bckdha*⁺/⁻ $n = 5$ *Bckdha*⁺/⁺ $n = 4$). **c** Leucine and alloisoleucine concentrations in plasma until age 6 months (*Bckdha*⁻/⁻ $n = 6$, *Bckdha*⁺/⁻ $n = 11$, *Bckdha*⁺/⁺ $n = 6$, all injected). One *Bckdha*⁻/⁻ male mouse died on day 28 and was not included for the points >day 28. All data are shown as mean ± SD. Source data are provided as a Source data file.

mRNA transcripts and protein according to promoter expression profiles as previously described (Fig. S9). Therefore, $3 \times 10^{13}$ vg/kg dose of AAV8-hAAT-h*BCKDHA* was quite effective until 3 weeks of age, but not beyond, whereas the same dose of AAV8-EF1α-h*BCKDHA* showed more prolonged efficacy albeit with pup loss between 5 and 12 weeks of age.

Next, we tested the efficacy of $10^{14}$ vg/kg (high dose) AAV8-hAAT-h*BCKDHA* gene therapy. We injected six *Bckdha*⁻/⁻ mice and their littermates. One *Bckdha*⁻/⁻ mouse died on day 28 but the 5 other treated *Bckdha*⁻/⁻ animals exhibited long survival (>5 months), similar to mice treated with a high-dose AAV8-EF1α-h*BCKDHA* vector, albeit with a partial rescue of the remaining MSUD phenotype: lower weight and high plasma leucine concentrations (much higher than in the reciprocal experiment using $10^{14}$ vg/kg AAV8-EF1α-h*BCKDHA*) (Figs. 5a–c and S10).

BCKD enzyme activity was tested in the liver of *Bckdha*⁻/⁻ mice either untreated at age P2-4 or treated and sacrificed at age

one and/or six months (Fig. S11). At age P2-4, enzyme activity in untreated *Bckdha*⁻/⁻ mice was undetectable, contrasting with enzyme activity in litter-matched controls that was comparable or slightly reduced relative to older mice. Compared to age-matched controls, at age 6 months, *Bckdha*⁻/⁻ mice treated with the AAV8-EF1α-h*BCKDHA* or the AAV8-hAAT-h*BCKDHA* vectors at $10^{14}$ vg/kg showed 12 ± 4% and 7 ± 4.5% BCKD activity, respectively. Of note in AAV8-EF1α-h*BCKDHA*-treated mice at $10^{14}$ vg/kg, the activity was much greater at age one month than at age 6 months, though with high variance, consistent with progressive vector dilution in the growing liver (Fig. S11). In these two situations, however, the degrees of reconstituted hepatic BCKD activity in the *Bckdha*⁻/⁻ mice were markedly lower than the restoration of the hBCKDHA protein to the wild-type levels in the livers (Figs. 3e and 4f). In AAV8-EF1α-h*BCKDHA*-treated mice at $10^{13}$ vg/kg that were only transiently and partly rescued, activity was reduced already

markedly at age one month, i.e., $7 \pm 5\%$ activity relative to age-matched controls (Fig. S11).

Finally, as the two cassettes (EF1α and hAAT) differed in one intron (iEF1α vs. iSV40, respectively), we wanted to confirm in vitro data showing that in Huh7 cells, there was no overt difference in terms of BCKDHA protein expression between the original hAAT driven construct containing the SV40 intron, iSV40 and an additional hAAT driven construct containing the EF1α intron, iEF1α. We therefore compared outcomes of intermediate dose ($3.10^{13}$ vg/kg) injected individuals with these two constructs in a separate experiment. We injected 3 $Bckdha^{-/-}$ individuals immediately after birth with the AAV8-hAAT-iSV40-h$BCKDHA$ construct, which showed neurological symptoms during the fourth week of life, 2 of them with weight loss, requiring sacrifices before age 4 weeks. Six $Bckdha^{-/-}$ individuals were injected with the AAV8- hAAT-iEF1α-h$BCKDHA$ construct. Two showed neurological symptoms with weight loss during the fourth week of life requiring sacrifices before age 4 weeks. Three exhibited a major growth arrest during the fourth week of age. All of these mice were sacrificed at age 4 weeks. In keeping with the in vitro data (Fig. S2a), no major differences were found between the two constructs considering growth and survival, plasma leucine concentrations, and BCKDHA protein expression in the liver (Fig. S2). A non-significant trend towards superior rescue by AAV8-hAAT-iEF1α-h$BCKDHA$ cannot be excluded but any such effect falls within interindividual variability.

These data may suggest a significant contribution of non-liver tissues to the whole-body BCKDHA transgene activity responsible for the phenotypic rescue of MSUD in neonate mice as they show that at equivalent vector doses, a better phenotypic rescue is achieved with the EF1α-driven AAV8 transgene.

## Discussion

MSUD represents an unmet medical need with limited therapeutic curative options. Given the beneficial impact of liver transplantation on survival, quality of life, and disease biomarkers in patients with MSUD, we hypothesized that liver-directed gene therapy would be a feasible therapeutic approach. Herein, we showed that a single AAV vector infusion in neonate MSUD mice resulted in the long-term survival of animals and rescue of the disease phenotype.

As mutations in $BCKDHA$ are one of the most frequent causes of MSUD in humans[2], a first step was to establish and characterize a colony of $Bckdha^{-/-}$ mice. We showed that homozygous $Bckdha^{-/-}$ mice faithfully recapitulate the human disease features, with affected animals exhibiting extremely high plasma concentrations of leucine and early-onset severe neurological involvement resulting in death before age P12.

Based on the clinical experience, heterozygous human individuals, like $Bckdha^{+/-}$ mice, exhibit 50% of whole-body BCKD enzyme activity and do not exhibit any phenotypic or biochemical abnormalities. This would suggest that the restoration of 50% of BCKD activity could be fully curative. However, liver transplantation in humans brings approximately 10% of the whole-body BCKD activity[10] and dramatically improves the MSUD phenotype and the metabolic parameters for the great majority of patients[4,28]. Thus, based on these considerations, in the context of gene therapy, an increase of BCKDH above 10% of normal would be sufficient to rescue the disease phenotype, while levels around 50% of normal would completely correct MSUD. Emerging data from liver-directed gene therapy trials with AAV vectors suggest that levels of transduction needed to rescue MSUD are indeed achievable, as in some cases supraphysiological levels of transgene expression have been achieved in hemophilia A subjects[29], and

whole-body transduction has been documented in infants affected by spinal muscular atrophy[30] and Duchenne muscular dystrophy[31], among others. One of the challenges of AAV vector-mediated liver gene transfer is that, due to the predominantly non-integrating nature of these vectors, transgene expression in rapidly dividing tissues is not stable. This has been shown in several preclinical animal models[32–34] in which AAV vectors have been administered at a time when the liver had not reached the adult size. This is clearly a limitation in the development of neonatal gene therapy for MSUD, as early treatment, ideally shortly after birth, would greatly benefit patients.

Here, by using the ubiquitous promoter EF1α, we were able to achieve full and sustainable phenotypic rescue both clinically and biochemically in neonate $Bckdha^{-/-}$ mice. BCKDHA transgene mRNA and protein were detectable in the liver, brain, and heart, reflecting the widespread transduction of the vector given at a dose of $10^{14}$ vg/kg, in line with doses tested in neonatal patients[30]. At a lower vector dose of $10^{13}$ vg/kg, the rescue was partial and transient, indicating subtherapeutic levels of transgene expression. Future studies aiming at optimizing the transgene expression cassette[34], testing of hybrid promoters[35], and possibly testing vectors with broader tropism than AAV8[31], will help lowering the therapeutic vector dose threshold.

By contrast, liver-restricted gene transfer using the hAAT promoter[36] provided partial though sustained rescue of the MSUD phenotype. Indeed, mice injected with vector doses of $10^{14}$ vg/kg exhibited lower weight growth, plasma leucine concentrations of about 4.5–6 times the normal (vs. 2 times in the ubiquitous approach). This suggests that the lack of extrahepatic expression may reduce therapeutic efficacy, thus consistent with previous findings from our laboratory, showing that multi-tissue expression of alpha-acid glucosidase provides superior rescue of Pompe disease in neonate mice compared to liver-restricted transgene expression[35]. An alternative, possibly complementary mechanism suggested by BCKD activity assays is that the hAAT construct may be less active than the EF1α construct in the liver in vivo, though previous reports showed that the hAAT promoter is more potent than the EF1α one[37].

Our data suggesting that in the setting of neonatal gene therapy, long-term disease rescue with the ubiquitous promoter may involve extra-hepatic correction of the molecular defect, are consistent with recent data[38] suggesting that similar to humans, branched-chain amino acid oxidation takes place in most mice tissues including, muscle, brown fat, liver, kidney and heart. In keeping with this, recent findings show disease rescue in a mild mouse model of MSUD (intermediate MSUD [iMSUD] mouse model) by AAV9 vector expressing E2 not solely in the liver but also in muscle at a non-neonatal age[39]. This model harbors a mutation in the E2 [$DBT$] subunit of BCKDH and has a phenotype that is milder than the classical form of MSUD explaining why it could be rescued by systemic injection at weaning (P21-P28, equivalent to late adolescence in humans). Relative to ubiquitous expression at the highest vector doses tested, liver-specific gene transfer leads to long-term survival but poorer growth and higher concentrations of metabolic parameters. Indeed, in the mouse model used in our study, animals had to be treated immediately at birth in order to rescue the early lethality associated with MSUD. As discussed above, pronounced liver growth during the first weeks of life in mice[32,40] results in the dilution of the AAV vector genomes and loss of transgene expression possibly explaining the partial efficacy of the liver-specific (hAAT) approach. Additionally, the fact that restoration of BCKDHA expression in the liver to supraphysiological levels is not fully correcting the disease can also be due to several other factors. These include the expression of the enzyme in only a subset of hepatocytes. Based on literature data[41] and our own experience

with AAV vectors, it is safe to assume that at the doses used herein, nearly 100% of mouse hepatocytes would express the enzyme. In large animal models, the efficiency of AAV transduction in the liver is significantly lower[42] as it is also expected to be in humans. The lack of sufficient expression in other tissues, or the production of a non-functional and/or misfolded BCKD enzyme in hepatocytes due to overexpression may also play a role. The recombinantly expressed E1 protein has been shown to depend on molecular chaperones for proper folding and heterotetrameric assembly in cells[26,43,44].

Based on these observations, and given that in humans the liver reaches nearly full size by the age of 10 years[45], it is reasonable to argue that in the context of MSUD and other similar diseases with early manifestations, strategies to achieve long-term efficacy following AAV gene transfer are highly needed. As an alternative to this approach and a general strategy to achieve the long-lasting rescue of pediatric diseases with primary liver manifestation[46] with hepatocyte-directed gene transfer, multiple vector administrations over time should be envisioned. The development of humoral immunity against the infused vector post gene transfer[47] represents one of the most important limitations of vector redosing[48,49]. Recent developments on the use of immunosuppressive agents to prevent anti-AAV vector antibody formation[50,51], and on the use of IgG degrading enzymes[52] and other methods aimed at removing antibodies[53–55], will likely address the limitation of anti-AAV neutralizing antibodies moving forward. Alternatively, the use of mRNA-based therapies[56] may help bridge patients to an age in which the liver is fully formed when a single AAV vector infusion would provide a long-lasting therapeutic effect.

In conclusion, we provide strong evidence of the long-term therapeutic efficacy of AAV8 neonatal gene therapy in a mouse model of MSUD that is lethal shortly after birth. Further research is warranted to precisely evaluate the potential contribution of extrahepatic tissues to disease rescue as opposed to dose-dependent effects within the liver. This study is of interest for a broader subset of genetic conditions characterized by early, postnatal onset and for which both liver and peripheral tissue expression could rescue the disease long term.

## Methods

**Plasmid design and vector production.** The h*BCKDHA* gene (NCBI gene ID: 593) comprises 9 coding exons, spanning around 27 kb on chromosome 19, resulting in a cDNA of 1.34 kb. The *BCKDHA* transgene expression cassettes were WT versions of the human *BCKDHA* coding sequence, encoding the full-length hBCKDHA protein. We designed 2 transgene expression cassettes. The first cassette, termed EF1α-h*BCKDHA* was assembled by cloning the h*BCKDHA* cDNA into a modified AAV expression cassette that includes the wild-type AAV2 inverted terminal repeats (ITRs), and a ubiquitous human elongation factor 1-alpha promoter (hEF1α) with an EF1α intronic sequence to increase the expression of the transgene. To improve mRNA stability, we also included the Woodchuck Hepatitis Virus (WHP) Posttranscriptional Regulatory Element (WPRE), and the bovine growth hormone gene polyadenylation signal (bGH poly(A)). The second cassette, termed hAAT-h*BCKDHA* assembled the h*BCKDHA* cDNA, the AAV2 ITRs, a hepatocyte-specific hAAT promoter composed of the ApoE enhancer and the human *alpha 1 antitrypsin promoter* (hAAT)[57] and an SV40 intronic sequence or an EF1α intronic sequence, the WPRE sequence and the polyadenylation signal (bGH ploy(A)) to improve mRNA stability. All DNA sequences used in this study were synthesized by GeneCust. We also designed two codon-optimized h*BCKDHA* transgenes under the control of the EF1α or hAAT promoters. In total, there were 6 transgenes expression cassettes: 3 for each promoter: the full-length human *BCKDHA* cDNA sequence and two codon-optimized sequences (co1 and co2). The AAV vectors were produced using an adenovirus-free transient transfection method of HEK293 cells (HEK293 cell line (ATCC-USA), a kind gift from Dr. Agnès Rötig from Institut Imagine - Inserm UMR 1163, Paris, France) and purified by CsCl gradient. Titers of AAV vector stocks were determined using real-time qPCR and SDS-PAGE, followed by SYPRO Ruby protein gel stain and band densitometry. All vector preparations used in the study were titered side by side before use. The primers used for qPCR on the AAV genome annealed to bGH polya signal region of the expression cassette sequence: forward: 5′-gccactcccactgtcctttc-3′; reverse, 5′-cccagcatgcctgctattgt-3′. The

serotype used was AAV8, which is known to target the liver and has been tested in the clinic for the treatment of pediatric disorders.

**Assessment of transduction capacity in vitro.** Human hepatoma cells (Huh7 (ATCC-USA), a kind gift from Pr. Jessica Zucman-Rossi from Centre de Recherche des Cordeliers – Inserm UMR S1138, Paris, France), were seeded in 6-well plates ($5 \times 10^5$ cells/well) and transfected with the plasmids to be tested (2 μg/well) using Lipofectamine 2000 in OPTIMEM medium (Thermo Fisher Scientific) according to the manufacturer's instructions. A plasmid encoding GFP under the control of EF1α or hAAT promoters (2 μg/well) was transfected as a control. Seventy-two hours after transfection, cells and conditioned media were harvested and analyzed for hBCKDHA expression by western blot.

**Animal study procedures.** Mouse studies were performed according to the French and European legislation regarding animal care and experimentation (2010/63/EU) and approved by the local institutional ethical committee (Paris Descartes ethics committee CEEA34, APAFIS#22754-2018092017287553 v3). *Bckdha*[+/−] mice were purchased from The Canadian Mouse Mutant repository (C57BL / 6N-Bckdhae-m1(IMPC)Tcp). These mice carried a 266-bp deletion of Chr7 from 25638173 to 25638438 and insAGAGCC at the heterozygous state and *Bckdha*[−/−] mice have never been characterized.

*Bckdha*[+/−] mice were crossed to generate *Bckdha*[−/−] mice. To evaluate the therapeutic efficacy of our constructs, we performed single systemic intra-temporal injections of AAV8-EF1α or hAAT h*BCKDHA* transgene vectors in neonate mice at P0. The volume of injection was 10 μL. We injected the whole litters and genotyped the pups at P7. Different doses were tested: $10^{13}$ vg/kg, $3 \times 10^{13}$ vg/kg, and $10^{14}$ vg/kg. Mice were housed in a temperature (20–22 °C) and humidity (40–50%)-controlled environment with 12 h/12 h light/dark cycle and fed ad libitum with a standard diet.

Blood sampling was performed every 2 weeks and at sacrifice. Plasma was separated by centrifugation at 1000 g for 10 min and stored at −80 °C until analysis. After sacrifice, organs were snap-frozen in liquid nitrogen and stored at −80 °C for further analysis or fixed overnight in 4% paraformaldehyde for pathology studies.

**Genotyping.** DNA extraction from the tail or ear clip was performed by adding 75 μL of extraction solution (25 mM NaOH, 2 mM EDTA) and by heating at 95 °C for 45 min. Samples were cooled on ice for 5 min. An equal volume of 40 mM Tris HCl (pH 7.5–8) was then added. DNA was amplified using a Taq DNA Polymerase PCR kit (Peqlab, Germany) according to the manufacturer's instructions. The first couple of primers, flanking the deleted region in *Bckdha* (forward: 5′-ctgtccacttccctttggag-3′; reverse: 5′-ctttccatgctggttcttgg-3′), were used, giving either a 516 bp amplicon for the wild-type allele (without the deletion) or a 256 bp amplicon for the mutated allele (with the deletion). The second set of primers was used with the antisense primer comprised in the deleted region of *Bckdha* (forward: 5′-ctgtccacttccctttggag-3′; reverse: 5′-agttggtcatgtagaaggagatcc-3′), giving a 339 bp amplicon for the wild-type allele (without deletion), or no amplicon for the mutated allele (with the deletion). Amplification conditions were 95 °C for 10 min then 40 cycles of 95 °C for 30 s, 62 °C for 30 s, 72 °C for 1 min.

**Vector genome copy number analysis.** DNA was extracted from tissues homogenates using DNeasy Blood & Tissue kit (Qiagen, Germany) and quantified. Vector genome copy number was determined by qPCR using 100 ng of DNA (assay with Nanodrop, version 1.6.198). Primers annealed to bGH polya signal region of the expression cassette sequence: forward: 5′-gccactcccactgtcctttc-3′; reverse, 5′-cccagcatgcctgctattgt-3′. Data were recorded with QuantStudio (version 5.2).

**RNA extraction and expression.** Total RNA was extracted from snap-frozen tissue homogenates using the RNeasy mini kit (Qiagen, Germany). RNA was quantified and 0.8 mg (quantified with Nanodrop, version 1.6.198) was retro-transcribed to cDNA using the QuantiTect Reverse Transcription Kit (Qiagen, Germany). RT-minus reactions were performed as a negative control. For h*BCKDHA* mRNA and mice *Bckdha* expression analyses, the qPCR on cDNA was performed using Sybergreen and primers annealing specifically on the WT h*BCKDHA* transgene sequence (h*BCKDHA* forward, 5′-cccacctctgagcagtatcgc-3′; h*BCKDHA* reverse, 5′-caaacacatcattaccatccacgc-3′) and the mice *Bckdha* sequence respectively (*Bckdha* forward, 5′-gatgcctgttcactacggct-3′, *Bckdha* reverse, 5′-cacaatccggttggcatggg-3′). Mouse *Gapdh* gene transcripts were used to normalize *Bckdha* or h*BCKDHA* expression (*Gapdh* forward, 5′-ggcattgtggaagggctcat-3′; *Gapdh* reverse, 5′-gtcttctgggtggcagtgat-3′). Data were recorded with QuantStudio (version 5.2).

**Western blot.** Snap frozen mouse tissues were weighted, homogenized with RIPA buffer supplemented with protease inhibitors, sonicated using the Bioruptor system (Diagenode, Belgium), and centrifuged 20 min at 20,000 × *g* to collect the supernatant. Protein concentration was determined using the BCA protein assay (Thermo Fisher Scientific). SDS-PAGE electrophoresis was performed in a 4–12%

polyacrylamide gel. After transfer, the membrane was blocked with a solution of 5% BSA and incubated with an anti-BCKDHA antibody (Rabbit polyclonal; Abcam; cat. no. ab126173; lot GR126952; WB 1:1000) and anti-GAPDH (Mouse monoclonal; Clone 6C5; Abcam; cat. no. ab8245; lot GR3317834; WB 1:1.000). The membrane was washed and incubated with the appropriate secondary antibody (donkey anti-IgG Rabbit; Donkey polyclonal; LiCor; cat. no. 926 32213; lot C70918; WB 1:10,000 or donkey anti-IgG Mouse; Donkey polyclonal; LiCor; cat. no. 926 68072; lot D00226; WB 1:10,000) and visualized with the Odyssey imaging system (LI-COR Biosciences) and Image Studio version 5.2.

The polyclonal anti-BCKDHA antibody recognized both the mouse BCKDHA and human BCKDHA proteins. For quantification, the hBCKDHA and mouse BCKDHA protein expression was normalized to GAPDH or actin (anti-actin, mouse polyclonal; ProteinTech; cat. no. 20536-I-AP; WB 1:1.000) protein expression.

The same protocol was used for the detection and quantification of phosphorylated BCKDHA (Se293) using the Anti-human phosphoBCKDHA antibody (Rabbit polyclonal; Abcam; cat. no. ab200577; lot GR369441; WB 1:2000).

For BCKDHB detection, the polyclonal anti-BCKDHB antibody (Rabbit polyclonal; Abcam; cat. no. ab201225; lot GR3244895; WB 1:200) was used and protein expression was normalized to GAPDH protein expression.

Uncropped and unprocessed scans of western blots are provided in the Source data file.

**Biochemical characterization**. Plasma concentrations of amino acids were measured with liquid chromatography coupled to tandem mass spectrometry (UPLC-MS/MS). For accurate quantification, a stable isotope internal standard for the same structure for each metabolite (all purchased from Eurisotop, Saint Aubin, France) was added to the sample before protein precipitation. Samples were first derivatized using the AccQ Tag™ Ultra (Waters Corporation, Milford, MA, USA) according to manufacturer recommendations. Amino acid separation was performed with an Acquity™ UPLC system using a CORTECS™ UPLC C18 column (1.6 μm, $2.1 \times 150$ mm) coupled to microTQS™ tandem mass spectrometer operating with the software MassLinks (version 4.2, Waters Corporation, Milford, MA, USA). Internal labeled standards for quantification were labeled on all carbons and nitrogen(s).

Plasma levels of 2-ketoisocaproic (KIC) and 2-keto-3-methylvaleric (KMV) acids were measured with liquid chromatography coupled to tandem mass spectrometry (UPLC-MS/MS). A stable isotope internal standard for 2-ketoisocaproic acid labeled on all carbons (purchased from Eurisotop, Saint Aubin, France) was added to samples before protein precipitation by methanol. Underivatized samples were analyzed with a Nexera 30™ UPLC system, using an Astec CHIROBIOTIC™ column (5 μm × 30 cm × 2.1 mm) coupled to a Shimadzu 8060 triple quadrupole tandem mass spectrometer (Shimadzu Corporation, Kyoto, Japan). The acquisition was performed in a negative ionization mode with the MRM transitions 129.15 > 85.05 (precursor $m/z$ > product $m/z$) for natural KIC and KMV (KIC and KMV were not separated in this analysis and were measured together) and 135.15 > 90.10 for (M + 6)-labeled KIC. The KIC + KMV plasma level was calculated from the ratio between the sum of the corresponding signal areas and the signal area of the KIC M + 6 internal standard of a known quantity.

**Liver BCKD enzyme activity**. We performed an analysis of actual BCKD activity in the liver with a spectrophotometric assay as previously described[58], based on monitoring of NADH production and using 2-ketoisovaleric acid as substrate. As validated[59], we did not use α-chloroisocaproate in the extraction buffer and dihydrolipoamide dehydrogenase in the assay buffer. Approximately 25 mg of frozen liver was pulverized with a mortar and homogenized in 200 μL of ice-cold extraction buffer (50 mM HEPES, 3% (w/v) Triton X-100, 2 mM EDTA, 5 mM dithiothretiol (DTT), 0.5 mM thiamine pyrophosphate, 50 mM potassium fluoride, 2% (v/v) bovine serum, 0.1 mM $N$-tosyl-L-phenylalanine chloromethyl ketone (TPCK), trypsin inhibitor (0.1 mg/mL), leupeptin (0.02 mg/mL); pH 7.4 at 4 °C), adjusted with KOH). Centrifugation at $20,000 \times g$ for 5 min at 4 °C removed insoluble materials. The supernatant was diluted to 9% PEG, stood on ice for 20 min, and was centrifuged at $12,000 \times g$ for 10 min at 4 °C. The pellet was suspended in 200 μL of ice-cold suspending buffer (25 mM HEPES, 0.1% (w/v) Triton X-100, 0.2 mM EDTA, 0.4 mM thiamine pyrophosphate, 50 mM KCl, and leupeptin (0.02 mg/mL); pH 7.4 at 37 °C, adjusted with KOH). 500 μL of assay buffer (60 mM potassium phosphate dibasic, 4 mM MgCl2, 0.8 mM thiamine pyrophosphate, 0.8 mM CoA, 2 mM NAD + 0.2% (w/v) Triton X-100, 4 mM DTT), 460 μL of $H_2O$ and 20 μL of tissue extract were mixed. Enzymatic activity measurement was performed at 30 °C after the addition of 20 μL of 50 mM 2-ketoisovaleric acid (final concentration 1 mM; pH of the final assay = 7) by monitoring NADH production at 340 nm (with the software Kinetics version 3.00). Protein concentrations of the tissue extract were assessed with a BCA assay. The activity was normalized to proteins and expressed in arbitrary units.

**Pathology studies**. Organs were fixed overnight in 4% paraformaldehyde. For conventional bright-field light microscopy, organs were embedded in paraffin. Sections were stained with hematoxylin and Eosin. Numerisation of sections was performed with the software NDP.view2 (version 2.7.41).

**Statistical analysis**. All the data shown are reported as means ± standard deviation (SD). The statistical analyses were performed using the R software (version 3.4.2, http://www.R-project.org) and GraphPad Prism (version 9.3.1). Non-parametric tests were performed when two groups were compared (two-sided Mann–Whitney test). The survival of $Bckdha^{-/-}$ mice was compared to the survival of $Bckdha^{+/-}$ or $Bckdha^{+/+}$ mice by log-rank Mantel–Cox test.

**Reporting summary**. Further information on research design is available in the Nature Research Reporting Summary linked to this article.

## Data availability

All data generated or analyzed during this study are included in this published article (and its Supplementary Information files). Source data are provided with this paper.

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

## Acknowledgements

This work was supported by "DIM" (*Domaine d'Intérêt Majeur*) of the Paris Ile-de-France Region grant to C.P., M.S.-S., V.G., C.G., A.R., C.O., M.C., and M.S. The authors are indebted to the AAV platform and the animal facility of Imagine Institute, the histology and microscopy platforms of Institut Necker Enfants Malades, the mass spectrometry platform, and the Centre d'Investigation Clinique of the Biotherapy Department, Necker Hospital. We thank Mrs Aurore Besse, Aline Huguet, and Hélène Benyamine for their help in training for injections. We also thank Mrs. Dominique Farabos and Dr. Antonin Lamazière (Saint-Antoine University Hospital, Paris, mass spectrometry platform).

## Author contributions

C.P. and M.S. designed and performed the experiments, analyzed the data, and wrote the manuscript. M.S.-S., C.G., V.N., I.R., and N.T. performed the experiments, analyzed the data, and revised the manuscript. P.C., M.G., M.-G.B., C.O., F.M., and M.C. designed the experiments, analyzed the data, and revised the manuscript. J.-B.A., A.R., and P.d.L. revised the manuscript.

## Competing interests

F.M. and M.-G.B. are employees and equity holders of Spark Therapeutics, a Roche company. We declare that C.P., M.S.-S., P.d.L., C.O., M.C., and M.S. are designated as inventors of the European Patent Application EP20305079.4 filed on Jan. 29, 2020 in the names of Inserm, Fondation Imagine, Université de Paris, and APHP for "Gene therapy for MSUD." The remaining authors declare no competing interests.
