## [Peer Review File · Nature Communications]

Reviewer #1 (Remarks to the Author):

In the manuscript by Pontoizeau, the generation of a model of maple syrup urine disease (MSUD) is described based on the deficiency of the Bckdha protein, furthermore the authors developed an AAV8 vector for human Bckdha gene delivery to correct the disease. MSUD is a devastating disease with a high unmet medical need with no appropriate treatments available except for liver transplantation. In the present work, the authors have shown that Bckdha^{-/-} mice show a very severe phenotype with lethal early onset, neurological impairment, and alloisoleucine accumulation. The authors tested two different expression cassettes, in one the expression of the therapeutic protein is controlled by an oblique expression promoter (EF1a) and the second one by a liver specific promoter (ApoE Enh AATp). The authors were able to significantly ameliorate the disease phenotype and rescue mice from an early death after administration of a high dose of the oblique expression vector at birth, establishing the proof of concept for the development of an AAV therapeutic vector for the MSDU.

The authors conclude that disease correction requires the expression of the protein in other organs apart from the liver since the therapeutic effect was achieved with a high dose ubiquitous expression vector but not with the liver specific. This finding is surprising since in patients the disease can be corrected but liver transplantation and in mice in 80% of body BCKD activity come from the liver. The results are of interest and establish the basis for the development of a therapeutic vector for this disease, however, the authors have not performed the proper experiment to conclude that oblique expression is required and they are a number of aspects of the manuscript that should be improved:

The expression cassettes in the EF1a driven construct is different from the ApoE-AAT construct, the first one contains the EF1a intron and the second one the SV40 intron. According to data presented in Sup fig1 the expression of the protein using the EF1a is stronger than the one obtained with ApoE-AAT. Knowing that, the in vitro data are only indicative and it doesn't have to strictly reflect the expression levels in vivo in mouse hepatocytes, the reviewer believes that to claim that oblique expression of Bckdha is required to achieve a therapeutic and liver restricted expression is not sufficient a more fair comparison is needed. The two constructs should carry the same expression cassette except for the promoter and the same dose of vector should be administered.

In the results section regarding the selection of the plasmids very limited information has been included and the heading of the paragraph is not correct since the authors are testing the AAV transfer plasmid and not the AAV vectors. More information about the different plasmids tested should be included and western blot analysis now in supplementary should be included in the main figure 2 since it is not a very busy figure.

The figures should be improved and more information to identify the groups should be included, vector administered and dose. In the bar graphs, individual data should be included in all the figures. Figure 3: additional animals treated with a dose of 1014 µg/kg have been included in figure 1B but there is no information about the origin of those animals, it says exp 1 but there is no information about that experiment.

What does it mean that in mice treated with 1014 µg/kg they have subnormal plasma leucine concentration? Sub-normal in my understanding means lower than normal but this is not the case since the levels are still higher than normal.

Figure 4, panels G and H are difficult to understand, in red should be the animals receiving the vector after 12 months but according to figure H those animals are death.

To improve the study it will be interesting to include Isolleucine values in serum monthly and not at a single time point particularly in the long-term study. Furthermore, since the protein is expressed over normal levels in the liver, heart and brain the histology of those organs should be included.

Figure 5. In my opinion instead of including the data with the low dose of vector it will be more informative to include the data obtained with the dose of 3×10^{13} of the two vectors, showing genome, mRNA and protein levels.

In panel E (fig 5), It is surprising to see protein expression in the heart of KO mice at levels similar to WT animals, how the authors explain this finding.

Discussion

It is true that 50% of BCKGH activity is enough to avoid allosleucine accumulation and disease pathology since heterozygous animals are normal, and the authors claim that restoring 50% of the activity in the liver will result in the correction of the disease. However, it is doesn't seem to be the case since as shown in figure 5E protein levels are higher than normal at least twice and the disease was not corrected. The difference between and heterozygous and a vector treated mice will be most likely the percentage of cells expressing the protein, in the case of the heterozygous it will be 100% and the information about how many cells are transduced with the vector is missing this information is very important and should be include and discussed in the paper.

Reviewer #2 (Remarks to the Author):

The manuscript by Pontoizeau et al. describes AAV-mediated neonate gene therapy of maple syrup urine disease (MSUD), a rare severe inherited disorder in branched-chain amino acid catabolism. The authors employed both a ubiquitous and a liver-specific promotor to drive expression of the BCKDHA gene in a *bckdha*^{-/-} mouse model. The use of the ubiquitous EF1 α promoter results in a full and long-term correction of MSUD. In contrast, the liver-specific expression of BCKDHA driven by the hAAT promoter led to only a partial transient rescue of the metabolic disease.

Strengths:

- The manuscript is well-written; the results are straightforward and in general clearly presented. The correction of MSUD via the systemic expression of BCKDHA is a positive outcome, based on the long-term survival, reduced plasma leucine levels and absence of brain neuropathology.
- The work has a potential for clinical translation.

Weaknesses:

- The work is largely a technical feasibility study and lacks originality. The AAV-mediated gene therapy has been clinically used to correct a good number of other inherited diseases such as hemophilia A,

Duchenne muscular dystrophy and others. Along these lines, the present results do not support the conclusion that liver-specific expression of BCKDHA is associated with a partial transient rescue of MSUD. Rather, it is an artifact due to the instability of the AAV vector in rapidly dividing tissues such as the neonate *bckdha*^{-/-} mouse liver. As pointed out by the authors, orthotopic liver transplantations have led to significant long-term restoration of hepatic BCKD activity with a nearly complete reversal of MSUD phenotype. The latter findings support the prevailing concept that that liver is a major organ for branched-chain α -ketoacid (BCKA) disposal and is the target organ for MSUD gene therapy. To test the efficacy of hepatic gene therapy for MSUD, stable vectors other than the AAV should be used.

- Another major concern is the lack of functional characterization of restored BCKD function in *bckdha*^{-/-} mouse tissues. Only similar limited studies on BCKDHA mRNA and subunit levels are carried out in tissues transfected with both the ubiquitous (EF1 α -hBCKDHA) and the liver-specific (hAAT-hBCKDHA) AAV vectors. Significantly, no BCKD activity was determined in wild-type and transfected tissues. The phosphorylation state of the transfected BCKDHA subunit in tissues or the lack of it in liver is not shown. The BCKDHA subunit alone does not possess enzyme activity, therefore, the effects of the transfected BCKDHA on the expression of other BCKS subunits should be determined. For example, the BCKDHB subunit is absent in BCKDHA^{-/-} Mennonite classic MSUD cells (Koyata et al. Biochem. J. 1993). It is important to show that the stable or transient expression of the transfected *bckdha* gene restores the steady-state BCKDHB level, allowing assembly of the E1 protein and subsequently the functional BCKD complex. Without these and other functional insights, the manuscript is largely descriptive and premature for publication.

Additional Comments:

- Reduced plasma leucine concentrations alone are indirect to indicate restored BCAA catabolism. Concentrations of other BCAAs i.e. valine and isoleucine as well as BCKAs need to be determined. A complete amino acid analysis would indicate the specific reduction in circulating BCAA levels.
- Fig. S1 shows that there are two versions of the ubiquitous EF1 α -BCKDHA vectors: co1 and co2 (codon optimized sequences 1 and 2). However, it is not indicated which version (co1 or co2) was used in the short-term (Fig. 3) or the long-term (Fig. 4) study. Moreover, there were several missed AAV8 injections, resulting in deaths of *bckdha*^{-/-} mice, raising concerns regarding quality of the study. Finally, in Fig. S2, it is not clear why an “intermediate” dose (3x10¹³ vg/kg) instead of the effective high dose of the AAV8 vectors was injected.

REVIEWER COMMENTS

Reviewer #1 (Remarks to the Author):

In the manuscript by Pontoizeau, the generation of a model of maple syrup urine disease (MSUD) is described based on the deficiency of the Bckdha protein, furthermore the authors developed an AAV8 vector for human Bckdha gene delivery to correct the disease. MSUD is a devastating disease with a high-unmet medical need with no appropriate treatments available except for liver transplantation. In the present work, the authors have shown that Bckdha^{-/-} mice show a very severe phenotype with lethal early onset, neurological impairment, and alloisoleucine accumulation. The authors tested two different expression cassettes, in one the expression of the therapeutic protein is controlled by an ubiquitous expression promoter (EF1a) and the second one by a liver specific promoter (ApoE Enh AATp). The authors were able to significantly ameliorate the disease phenotype and rescue mice from an early death after administration of a high dose of the ubiquitous expression vector at birth, establishing the proof of concept for the development of an AAV therapeutic vector for the MSUD.

The authors conclude that disease correction requires the expression of the protein in other organs apart from the liver since the therapeutic effect was achieved with a high dose ubiquitous expression vector but no with the liver specific. This finding is surprising since in patients the disease can be corrected by liver transplantation and in mice in 80% of body BCKD activity come from the liver.

Response: We thank the reviewer for highlighting the relevance of our work and for helping us with the interpretation of the results presented. We agree that our findings may appear rather unexpected and surprising. Regardless, as cited in the discussion, recent *in vivo* data show that in mice, BCAA oxidation is distributed among several organs similar to humans (10% BCAA oxidation occurring in liver as opposed to historical *in vitro* data arguing that 80% BCAA oxidation took place in rodent liver)¹. Additionally, we extensively modified the discussion to address the reviewer's concern and included text on timing of injection, liver growth and subsequent AAV vector genome dilution. These three parameters should be taken into account for interpreting outcomes of liver-specific AAV gene transfer. Last, additional data provided in this revision and coming from a new set of experiments clearly demonstrated that equivalent "high doses" of AAV8 liver-specific gene therapy partially but not fully (as this is the case for the ubiquitous E1a cassette) rescued the disease phenotype (details below).

The results are of interest and establish the basis for the development a therapeutic vector for this disease, however, the authors have not performed the proper experiment to conclude that ubiquitous expression is required and they are a number of aspect of the manuscript that should be improved:

-The expression cassettes in the EF1a driven construct is different from the ApoE-AAT construct, the first one contains the EF1a intron and the second one the SV40 intron. According to data presented in Sup fig1 the expression of the protein using the EF1a is stronger than the one obtained with ApoE-AAT. Knowing that, the *in vitro* data are only indicative and it does not have to strictly reflect the expression levels *in vivo* in mouse hepatocytes, the reviewer believe that to claim that ubiquitous expression of Bckdha is required to achieve a therapeutic and liver restricted expression is not sufficient a more fair comparison is needed. The two

constructs should carry the same expression cassette except for the promoter and the same dose of vector should be administered.

Response: We thank the reviewer for this insightful comment. We have performed *in vitro* experiments comparing hAAT-iSV40-hBCKDHA (intron SV40 present in the initial hAAT construct) and hAAT-iEF1 α -hBCKDHA (new construct of the hAAT cassette where the intron SV40 was replaced by the EF1 α intron). We were able to show similar level of BCKDHA protein expression in transfected Huh7 cells between the two constructs (**Figure S2a**). Then, we compared outcomes of intermediate dose ($3 \cdot 10^{13}$ vg/kg) injected individuals with this novel cassette (hAAT iEF1 α) to the ones of individuals injected with the original construct (hAAT iSV40). In keeping with the *in vitro* data, no major differences were found. Similar findings for BCAA levels: comparable concentrations between iEF1 α and iSV40. This was added in the core text of the manuscript as well as in **Figure S2**. These results demonstrate validity of the comparison between the original ubiquitous and liver-specific constructs. Regarding administration of similar vector doses, as detailed below (see response to Reviewer 2), we performed a set of new experiments demonstrating that high dose (10^{14} vg/kg) liver-directed (hAAT) AAV8 gene therapy provided long-term sustained though partial disease rescue (**Figure S9**).

We would like to also point out that, regardless of differences in potency, the key difference between the EF1 α and hAAT promoter is tissue specificity. EF1 α is a ubiquitous promoter thus providing expression both in liver (lost over time with liver growth) and in other tissues. The hAAT promoter is expressed solely in liver, and expression is expected to be lost over time as a result of hepatocyte proliferation^{2,3}.

-In the results section regarding the selection of the plasmids very limited information has been included and the heading of the paragraph is not correct since the authors are testing the AAV transfer plasmid and not the AAV vectors. More information about the different plasmid tested should be included and western blot analysis now in supplementary should be included in the main figure 2 since it is not a very busy figure.

Response: We thank the reviewer for this helpful consideration. This has been corrected accordingly. We replaced “AAV vectors” in the heading of the paragraph by “cassettes”. As requested, Western blot data were included in the main **Figure 2**. We have accordingly specified that codon optimized constructs were not used as wild type constructs were those with highest BCKDHA expression.

-The figures should be improved and more information to identify the groups should be included, vector administered and dose. In the bar graphs, individual data should be included in all the figures.

Response: Figures were entirely revised and improved as requested.

-Figure 3: additional animals treated with a dose of 10^{14} vg/kg have been included in figure 1B but there is no information about the origin of those animals, it says exp 1 but there is no information about that experiment.

Response: We thank the reviewer for bringing this to our attention. “Experience 1” was referring to leucine plasma concentrations at age 4 weeks from mice which had been sacrificed at age 6 months. Complete results regarding these 6-month-old mice are now presented in the

paper. We agree that such way of reporting results was rather confusing. Therefore, we opted for not presenting these data on leucine concentrations at age 6 months in the main figure. These results are now presented in **Figure S4** showing leucine concentrations over time.

-What means that in mice treated with 1014 vg/kg they are subnormal plasma leucine concentration? Subnormal in my understanding means lower than normal but this is not the case since the levels are still higher than normal.

Response: Indeed, subnormal is not the appropriate word. We changed it to a proper characterization: “slightly higher than the upper limit of the normal range”.

-Figure 4, panels G and H are difficult to understand, in red should be the animals receiving the vector after 12 months but according to figure H those animals are dead.

Response: For clarification, we presented in **Figure 4** results for mice injected at 10^{14} vg/kg, collected during the first 6 months of life and at sacrifice at age 6 months. We added a new supplementary figure for the subset of mice from the same experiments who were kept alive until age 12 months (**Figure S6**). **Figure S6** shows extensive characterization of 12-month old treated mice (extension of weight curves, biochemistry, vector genome copy numbers, Western blot at sacrifice). These results were not presented in the initial version of the paper.

-To improve the study it will be interesting to include leucine values in serum monthly and not at a single time point particularly in the long-term study. Furthermore, since the protein is expressed over normal levels in the liver, heart and brain the histology of those organs should be included.

Response: As requested, we provided an additional figure (**Figure S4**) with longitudinal leucine and other BCAA and proteogenic amino acids plasma concentrations in the long-term study. Regarding histology, brain histology was already provided in **Figure S3**. We added liver pathology (**Figure S3 d, e**) showing no overt liver abnormalities in injected (WT and KO) mice. Unfortunately, heart histopathology evaluation was not performed.

Figure 5. In my opinion instead of including the data with the low dose of vector it will be more informative to include the data obtained with the dose of 3×10^{13} of the two vectors, showing genome, mRNA and protein levels.

Response: This was done as requested. In the new **Figure 5**, we presented weight, plasma leucine concentrations for the two constructs at 3×10^{13} vg/kg. In a supplementary figure (**Figure S8**), we presented vector genome copy numbers, mRNA and protein levels.

In panel E (fig 5), It is surprising to see protein expression in the heart of KO mice at levels similar to WT animals, how the authors explain this finding.

Response: We agree with the reviewers that this is unexpected and was possibly due to a nonspecific band. Therefore, we have repeated the experiments with slightly different experimental conditions and could get unequivocal data now included in the revised version of the manuscript (**Figure S7**).

Discussion

It is true that 50% of BCKDH activity is enough to avoid alloisoleucine accumulation and disease pathology since heterozygous animals are normal, and the authors claim that restoring

50% of the activity in the liver will result in the correction of the disease. However, it doesn't seem to be the case since as shown in figure 5E protein levels are higher than normal at least twice and the disease was not corrected. The difference between heterozygous and a vector treated mice will be most likely the percentage of cells expressing the protein, in the case of the heterozygous it will be 100% and the information about how many cells are transduced with the vector is missing this information is very important and should be included and discussed in the paper.

Response: The reviewer raises an important point. The lack of disease manifestations in heterozygous animals may be due the fact that not only the liver but all tissues across the body exhibit 50% residual enzyme activity. The fact that the restoration of BCKDHA expression in the liver to supraphysiological levels is not fully correcting the disease can be due to several factors, including the expression of the enzyme in only a subset of cells in the liver, the lack of sufficient expression in other tissues, or the production of non-functional enzyme in hepatocytes due to overexpression. This point will be addressed in future studies. Importantly, our work suggests that, in the setting of gene transfer, the contribution of non-liver tissues transduced with the AAV-BCKDHA vector is important for the long-term rescue of the disease. The efficiency of transduction of AAV vectors in the liver of small and large animal models has been extensively studied. Based on the literature ⁴, and based on our own experience with AAV vectors, it is safe to assume that at the doses used nearly 100% of mouse hepatocytes would express the enzyme. In large animal models the efficiency of AAV transduction in liver is significantly lower ⁵ as it is also expected to be in humans. With an eye on the future clinical translation of the work presented in our manuscript, careful dose finding studies in large models of gene transfer will have to be performed to identify a suitable therapeutic dose.

We commented on the above points in the revised version of the discussion.

Reviewer #2 (Remarks to the Author):

The manuscript by Pontoizeau et al. describes AAV-mediated neonate gene therapy of maple syrup urine disease (MSUD), a rare severe inherited disorder in branched-chain amino acid catabolism. The authors employed both a ubiquitous and a liver-specific promoter to drive expression of the BCKDHA gene in a *bckdha*^{-/-} mouse model. The use of the ubiquitous EF1 α promoter results in a full and long-term correction of MSUD. In contrast, the liver-specific expression of BCKDHA driven by the hAAT promoter led to only a partial transient rescue of the metabolic disease.

Strengths:

- The manuscript is well-written; the results are straightforward and in general clearly presented. The correction of MSUD via the systemic expression of BCKDHA is a positive outcome, based on the long-term survival, reduced plasma leucine levels and absence of brain neuropathology.
- The work has a potential for clinical translation.

Response: We thank the reviewer for her/his positive comments on our work and manuscript.

Weaknesses:

- The work is largely a technical feasibility study and lacks originality. The AAV-mediated gene therapy has been clinically used to correct a good number of other inherited diseases such as hemophilia A, Duchenne muscular dystrophy and others. Along these lines, the present results do not support the conclusion that liver-specific expression of BCKDHA is associated with a partial transient rescue of MSUD. Rather, it is an artifact due to the instability of the AAV vector in rapidly dividing tissues such as the neonate *bckdha*^{-/-} mouse liver. As pointed out by the authors, orthotopic liver transplantations have led to significant long-term restoration of hepatic BCKD activity with a nearly complete reversal of MSUD phenotype. The latter findings support the prevailing concept that that liver is a major organ for branched-chain α -ketoacid (BCKA) disposal and is the target organ for MSUD gene therapy. To test the efficacy of hepatic gene therapy for MSUD, stable vectors other than the AAV should be used.

Response: We thank the reviewer for these insightful comments. MSUD is a severe paediatric disease with early onset, thus we believe a chief concern in the development of a gene therapy-based approach to the disease is durability. While we agree that the literature on the correction of various diseases is rather extensive, we believe that our work contributes significantly to the development of *in vivo* gene therapies for MSUD and other diseases characterized by early, postnatal onset and for which both liver and peripheral tissue expression can correct the disease long-term. MSUD is both a severe disease and an unmet medical need, the fact that very little has been published on gene therapy for MSUD also highlights that it is not an easy disease to correct. To this end, recent work published by Greig and colleagues ⁶ shows rescue of MSUD in a mild mouse model of the disease with systemic AAV9 gene transfer at a non-neonatal age. In our study, immediate therapeutic neonatal management of MSUD remains of clinical relevance with a potential of clinical translation. Regardless, neonatal timing of injection (required in our experimental setting due to rapid lethality of the knockout pups) represented a limitation of liver-specific gene therapy long-term efficacy. Here, we provide strong evidence of long-term therapeutic efficacy in a mouse model of MSUD that is lethal shortly after birth, recapitulating the classical severe form of the disease in humans (85-95% of MSUD cases ⁷). We also provide evidence that the use of a ubiquitous promoter expressing in liver and other tissues is better suited to rescue the disease long-term, likely by expressing the therapeutic transgene long-term in extrahepatic tissues. Our approach relies on the systemic administration of an AAV vector, an approach that has successfully been used in infants affected by diseases like spinal muscular atrophy (SMA, *onasemnogene abeparvovec* ⁸).

We do understand that from what is shown in our manuscript to the actual human translation a lot of work needs to be done, including safety studies and studies in large animal models.

To reflect the considerations above, we edited the discussion section of the manuscript to highlight the novelty of our work and the additional work needed for clinical translation of the presented results.

- Another major concern is the lack of functional characterization of restored BCKD function in *bckdha*^{-/-} mouse tissues. Only similar limited studies on BCKDHA mRNA and subunit levels are carried out in tissues transfected with both the ubiquitous (EF1 α -hBCKDHA) and the liver-specific (hAAT-hBCKDHA) AAV vectors. Significantly, no BCKD activity was determined in wild-type and transfected tissues. The phosphorylation state of the transfected BCKDHA subunit in tissues or the lack of it in liver is not shown. The BCKDHA subunit alone does not possess enzyme activity, therefore, the effects of the transfected BCKDHA on the expression

of other BCKS subunits should be determined. For example, the BCKDHB subunit is absent in BCKDHA^{-/-} Mennonite classic MSUD cells (Koyata et al. Biochem. J. 1993). It is important to show that the stable or transient expression of the transfected *bckdha* gene restores the steady-state BCKDHB level, allowing assembly of the E1 protein and subsequently the functional BCKD complex. Without these and other functional insights, the manuscript is largely descriptive and premature for publication.

Response: We thank the reviewer for these insightful remarks.

Western blots of *Bckdha*^{-/-} liver extracts with anti-BCKDHB antisera were performed. We were able to confirm the Koyata et al. data⁹ that is absence of BCKDHB in *Bckdha*^{-/-} liver extracts. In treated individuals (BCKDHA EF1A AAV8 10¹⁴ vg/kg), BCKDHB protein could be rescued in liver by about 15% at age 6 months and 12 months (**Figures S5 and S6**). As requested, we studied the phosphorylation status of BCKDHA in treated individuals. In liver, hBCKDHA protein was less phosphorylated in *Bckdha*^{-/-} treated mice than in *Bckdha*^{+/+} mice, suggestive of an important active BCKDHA fraction at age 6 months and 12 months (**Figures S5 and S6**).

We performed an *in vivo* metabolic tracer analysis as a surrogate of BCKDH enzyme activity, in mice treated with 10¹⁴ vg/kg hAAT (liver-specific) AAV8 gene therapy which suggested a partial restoration of the metabolic through the BCKDH. We added a related paragraph accordingly.

Additional Comments:

- Reduced plasma leucine concentrations alone are indirect to indicate restored BCAA catabolism. Concentrations of other BCAAs i.e. valine and isoleucine as well as BCKAs need to be determined. A complete amino acid analysis would indicate the specific reduction in circulating BCAA levels.

Response: A complete amino acids (including BCAAs and proteogenic amino acids) analysis was provided at various longitudinal time points as requested (**Figure S4**).

BCKAs analyses in untreated *Bckdha*^{-/-} mice and in *Bckdha*^{-/-} mice treated with the AAV8-EF1α-hBCKDHA vector and sacrificed at age 6 months were provided, showing reduction of BCKAs accumulation after treatment (**Figures S1 and S5**).

- Fig. S1 shows that there are two versions of the ubiquitous EF1α-BCKDHA vectors: co1 and co2 (codon optimized sequences 1 and 2). However, it is not indicated which version (co1 or co2) was used in the short-term (Fig. 3) or the long-term (Fig. 4) study.

Response: We selected the EF1α- and hAAT-hBCKDHA transgenes with the full-length human BCKDHA cDNA sequence (WT) as these constructs showed the highest expression of BCKDHA in Huh7 cells. The constructs co1 and co2 were not further used in this study. For clarification, we added the following sentence: “Accordingly, codon-optimised constructs were not further used in this study.”

Moreover, there were several missed AAV8 injections, resulting in deaths of *bckdha*^{-/-} mice, raising concerns regarding quality of the study. Finally, in Fig. S3, it is not clear why an “intermediate” dose (3x10¹³ vg/kg) instead of the effective high dose of the AAV8 vectors was injected.

Response: We thank the reviewer for this important concern. Regarding the missed injection, we agree with the reviewer that it may indicate poor technical skills. The text did not reflect the reality of experimental activities: only few animals were not injected correctly and were excluded from the study. The text was revised to reflect this specific point.

Regarding the high dose hAAT AAV8 vector, we agree with the reviewer that this was a central issue. We added a set of new experiments and data to the manuscript. These important experiments demonstrated that high dose (10^{14} vg/kg) liver-specific (hAAT) AAV8 gene therapy provided long-term sustained though partial disease rescue (Figure S9). Treated individuals exhibited long survival (up to 8 months) albeit with an intermediate MSUD phenotype: lower weight and high leucine concentrations (much higher than in the reciprocal experiment using 10^{14} vg/kg EF1 α).

References

1. Neinast, M. D. *et al.* Quantitative Analysis of the Whole-Body Metabolic Fate of Branched-Chain Amino Acids. *Cell Metab.* **29**, 417–429.e4 (2019).
2. Bortolussi, G. *et al.* Life-Long Correction of Hyperbilirubinemia with a Neonatal Liver-Specific AAV-Mediated Gene Transfer in a Lethal Mouse Model of Crigler–Najjar Syndrome. *Hum. Gene Ther.* **25**, 844–855 (2014).
3. Wang, L. *et al.* AAV8-mediated Hepatic Gene Transfer in Infant Rhesus Monkeys (*Macaca mulatta*). *Mol. Ther.* **19**, 2012–2020 (2011).
4. Lisowski, L. *et al.* Selection and evaluation of clinically relevant AAV variants in a xenograft liver model. *Nature* **506**, 382–386 (2014).
5. Nathwani, A. C. *et al.* Self-complementary adeno-associated virus vectors containing a novel liver-specific human factor IX expression cassette enable highly efficient transduction of murine and nonhuman primate liver. *Blood* **107**, 2653–2661 (2006).
6. Greig, J. A. *et al.* Muscle-directed AAV gene therapy rescues the maple syrup urine disease phenotype in a mouse model. *Mol. Genet. Metab.* (2021) doi:10.1016/j.ymgme.2021.08.003.
7. Strauss, K. A. *et al.* Branched-chain α -ketoacid dehydrogenase deficiency (maple syrup urine disease): Treatment, biomarkers, and outcomes. *Mol. Genet. Metab.* **129**, 193–206 (2020).
8. Day, J. W. *et al.* Onasemnogene abeparvovec gene therapy for symptomatic infantile-onset spinal muscular atrophy in patients with two copies of SMN2 (STRIVE): an open-label, single-arm, multicentre, phase 3 trial. *Lancet Neurol.* **20**, 284–293 (2021).
9. Koyata, H., Cox, R. P. & Chuang, D. T. Stable correction of maple syrup urine disease in cells from a Mennonite patient by retroviral-mediated gene transfer. *Biochem. J.* **295** (Pt 3), 635–639 (1993).

Reviewer #1 (Remarks to the Author):

Now with the new experiment using the vector carrying the hAAT promoter at the higher dose of 1×10^{14} vg/kg the authors have addressed all reviewer's comments, however, there are several points that should be addressed by the authors particularly since there are some contradictory statements and some mistakes that require attention:

1. Figure S9. The data shown in the figure correspond to animals treated with hAAT vector and not to the EF1a vector, isn't it?
2. In the discussion, the statement in Page 15 line 329 "By contrast, liver-restricted gene transfer using the hAAT promoter provided partial though sustained rescue of the MSUD phenotype" is in contradiction with the statement in line 345: "In our study, liver-specific gene transfer does not lead to long-term efficacy".
3. Furthermore, if the therapeutic efficacy of the vector with the hAAT promoter is partial the same holds true for the EF1a promoter, since the data presented in figure S4 show that Leu, Val or Ile levels are not normalized. The reduction of these biomarkers is indeed more important with the EF1a vector; but the dose response observed indicates that most likely by increasing the dose of the hAAT vector similar results to the ones obtained with the EF1a promoter will be obtained, which if confirmed will indicate that expression in other organs is not strictly required. This point should be discussed in the context that liver transplantation is curative for MSED patients.
4. The data with the highest dose seems to be more relevant than the ones obtained with the intermediate dose and thus it will be better to include in the main figures the data at the highest dose of both vectors than at intermediate dose.
5. In the result section, the paragraphs in which the two doses of the hAAT vector are used, should be rewritten since as it is the therapeutic effect of the hAAT vector seems to be anecdotal and mainly to demonstrate the partial restoration of metabolic flux. It will be of interest to see the results when using the EF1a promoter.

Reviewer #2 (Remarks to the Author):

The revised manuscript by Pontoizeau et al. is significantly improved. The authors have addressed most of this reviewer's criticisms. However, a major deficiency remains to be the omission of measurements for BCKD activity in various tissues particularly the liver from the AAV vector-treated BCKDHA^{-/-} mice. These results are essential to provide direct evidence that a functional BCKD complex, or the lack of, is assembled upon the expression of the hBCKDHA subunit from the transfected ubiquitous and the liver-specific AAV vectors. The results of *in vivo* metabolic tracer analysis presented in Table S1 of the revised manuscript are indirect and weak.

The BCKD enzyme assay has been well described in the Methods of Enzymology. If the radiolabeled BCKA substrate [1-¹⁴C]KIV or [1-¹⁴C]KIC is not available, the spectrophotometric assay by monitoring NADH production for actual BCKD activity (no activation with the phosphatase) in various tissues are straightforward and will suffice. A simple preparation of mitochondria from liver, kidneys and heart markedly removes interfering reactions during the spectrophotometric assay.

Minor comments:

An added citation of the following article in the Introduction would strengthen the biochemical and clinical aspects of MSUD:

Chaung DT, Shih VE. Maple syrup urine disease (branched-chain ketoaciduria). In: Scriver CR, Beaudet AL, Sly WS, Valle D. eds. *The Metabolic and Molecular Bases of Inherited Disease*, 8th ed. New York, NY. McGraw-Hill; 2001: 1971-2006.

RESPONSE TO REVIEWERS' COMMENTS

Reviewer #1 (Remarks to the Author): Now with the new experiment using the vector carrying the hAAT promoter at the higher dose of 1×10^{14} vg/kg the authors have addressed all reviewer's comments, however, there are several points that should be addressed by the authors particularly since there are some contradictory statements and some mistakes that require attention:

1. Figure S9. The data shown in the figure correspond to animals treated with hAAT vector and no to the EF1a vector, isn't it?

We thank the reviewer for this comment concerning supplemental **Figure S9**. The name of the vector under the weight curve is now corrected (AAV8-hAAT-hBCKDHA construct). As recommended in point 4 below, we now included **Figure S9** as a main figure, *i.e.*, new **Figure 5**.

2. In the discussion, the statement in Page 15 line 329 “By contrast, liver-restricted gene transfer using the hAAT promoter provided partial though sustained rescue of the MSUD phenotype” is in contradiction with the statement in line 345: “In our study, liver-specific gene transfer does not lead to long-term efficacy”.

We edited the text accordingly in order to avoid any ambiguity. We now replaced the statement “In our study, liver-specific gene transfer does not lead to long-term efficacy” by “Relative to ubiquitous expression at the highest vector doses tested, liver-specific gene transfer leads to long-term survival but poorer growth and higher metabolic parameters”.

3. Furthermore, if the therapeutic efficacy of the vector with the hAAT promoter is partial the same holds true for the EF1a promoter, since the data presented in figure S4 show that Leu, Val or ile levels are not normalize. The reduction of these biomarkers is indeed more important with the EF1a vector; but the dose response observed indicates that most likely by increasing the dose of the hAAT vector similar results to the ones obtained with the EF1a promoter will be obtained, which if confirmed will indicate that expression in other organs is not strictly required. This point should be discussed in the context that liver transplantation is curative for MSUD patients.

We thank the reviewer for this insightful comment. Indeed, it is likely that higher doses of the vector containing the hAAT promoter than the ones used in our experiments will possibly result in similar efficacy as the EF1a vector. Regardless, higher AAV vector doses in humans have a higher likelihood to induce liver toxicities, complement activation and other potential issues. The experience with transplant in MSUD patients supports the fact that liver expression can correct the disease, however liver transplant is expected to restore 100% of BCKD enzyme in liver, while gene therapy directed uniquely to the liver is unlikely to achieve 100% of wild type expression. In this setting, extra hepatic expression of BCKD could possibly be beneficial. Accordingly, we have now tuned down the claim that previously emphasized the potential contribution of extra-hepatic tissues to disease rescue by adding the following statement at the end of the Discussion section: “Further research is warranted to precisely evaluate the potential

contribution of extra-hepatic tissues to disease rescue as opposed to dose-dependent effects within the liver”.

4. The data with the highest dose seems to be more relevant than the ones obtained with the intermediate dose and thus it will be better to include in the main figures the data at the highest dose of both vectors than at intermediate dose.

As requested, we have now moved **Figure 5** (comparison of vectors at intermediate dose) out of the main figures and into Supplementary information as new **Figure S8**. Previous **Figure S9** referring to the characterisation of mice treated at high dose with the hAAT vectors is now moved to the main figures as “new” **Figure 5**, augmented with additional data from the now available more extended follow-up (weight curves and plasma BCAA concentrations up to age 6 months).

5. In the result section, the paragraphs in which the two doses of the hAAT vector are used, should be rewritten since as it is the therapeutic effect of the hAAT vector seems to be anecdotal and mainly to demonstrate the partial restoration of metabolic flux. It will be of interest to see the results when using the EF1 α promoter.

We agree that data presentation of the low and intermediate doses of the hAAT vector suggested a rather negative effect while also going into somewhat too much detail. We now condensed the details and rephrased the corresponding paragraphs in the Results section in order to provide a more objective view of the partial rescue provided by the hAAT vector. Rewriting led to some long sentences, yet we hope these remain fully explicit and clearly stated. The rephrased text starts with sentence “Compared with the AAV8-EF1 α -hBCKDHA low dose (see above)” and ends with “showed more prolonged efficacy albeit with pup loss between 5 and 12 weeks of age.”

Rather than performing *in vivo* metabolic tracer analysis for the EF1 α promoter, we have followed the recommendation from reviewer 2 (see below) to directly measure BCKD enzyme activity in liver for both promoters. Therefore, we removed the data on metabolic flux associated with the hAAT promoter and the corresponding Table S1.

Reviewer #2 (Remarks to the Author):

The revised manuscript by Pontoizeau et al. is significantly improved. The authors have addressed most of this reviewer’s criticisms. However, a major deficiency remains to be the omission of measurements for BCKD activity in various tissues particularly the liver from the AAV vector-treated BCKDHA $^{-/-}$ mice. These results are essential to provide direct evidence that a functional BCKD complex, or the lack of, is assembled upon the expression of the hBCKDHA subunit from the transfected ubiquitous and the liver-specific AAV vectors. The results of *in vivo* metabolic tracer analysis presented in Table S1 of the revised manuscript are indirect and weak. The BCKD enzyme assay has been well described in the Methods of Enzymology. If the radiolabeled BCKA substrate [1- 14 C]KIV or [1- 14 C]KIC is not available, the spectrophotometric assay by monitoring NADH production for actual BCKD activity (no activation with the phosphatase) in various tissues are straightforward and will suffice. A simple preparation of mitochondria from liver, kidneys and heart markedly removes interfering reactions during the spectrophotometric assay.

We thank the reviewer for this recommendation. As suggested, we performed determination of actual liver BCKD activity with the spectrophotometric assay by monitoring NADH production, adapted from Nakai *et al*¹. We determined BCKD activity in the liver of untreated *Bckdha*^{-/-} mice at birth (P2-4) and *Bckdha*^{-/-} mice treated with the AAV8-EF1 α -hBCKDHA vector at 10¹⁴ vg/kg (sacrificed at age one month or six months) or at 10¹³ vg/kg (sacrificed at age one month). Similar experiments were performed in *Bckdha*^{-/-} mice treated with the AAV8-hAAT-hBCKDHA vector at 10¹⁴ vg/kg (sacrificed at age six months). We compared these enzyme activities to BCKD activity in liver of age-matched untreated *Bckdha*^{+/+} mice. Of note, results for BCKD activity were consistent with RT-PCR and Western blot data, confirming the validity of our results.

These novel data are shown as a new **Figure S11** in Supplementary Information. We added the following paragraph in the Results section:

“Branched chain ketoacid dehydrogenase (BCKD) enzyme activity was tested in the liver of *Bckdha*^{-/-} mice either untreated at age P2-4 or treated and sacrificed at age one and/or six months (**Fig. S11**). At age P2-4, enzyme activity in untreated *Bckdha*^{-/-} mice was undetectable, contrasting with activity in litter-matched controls that was comparable or slightly reduced relative to older mice. Compared to aged-matched controls, at age 6 months, *Bckdha*^{-/-} mice treated with the AAV8-EF1 α -hBCKDHA or the AAV8-hAAT-hBCKDHA vectors at 10¹⁴ vg/kg showed 12 \pm 4 % and 7 \pm 4.5 % BCKD activity, respectively. Of note in AAV8-EF1 α -hBCKDHA-treated mice at 10¹⁴ vg/kg, the activity was much greater at age one month than at age 6 months, though with high variance, consistent with progressive vector dilution in the growing liver (**Fig. S11**). In AAV8-EF1 α -hBCKDHA-treated mice at 10¹³ vg/kg that were only transiently and partly rescued, activity was reduced already markedly at age one month, i.e., 7 \pm 5 % activity relative to aged-matched controls (**Fig. S11**). BCKD activity levels were consistent with RT-PCR and Western blot results.”

Last, we now removed the paragraph and corresponding Table S1 on the *in vivo* flux analysis due to redundancy with the BCKD activity determination.

Minor comments: An added citation of the following article in the Introduction would strengthen the biochemical and clinical aspects of MSUD: Chaung DT, Shih VE. Maple syrup urine disease (branched-chain ketoaciduria). In: Scriver CR, Beaudet AL, Sly WS, Valle D. eds. The Metabolic and Molecular Bases of Inherited Disease, 8th ed. New York, NY. McGraw-Hill; 2001: 1971-2006.

We added this citation in the Introduction section, as requested. We also added a second reference, published during this revision, adding some information on the follow-up of transplanted patients (new ref 28), which does not modify our conclusions.

References

1. Nakai, N., Kobayashi, R., Popov, K. M., Harris, R. A. & Shimomura, Y. Determination of branched-chain alpha-keto acid dehydrogenase activity state and branched-chain alpha-keto acid dehydrogenase kinase activity and protein in mammalian tissues. *Methods Enzymol.* **324**, 48–62 (2000).

REVIEWERS' COMMENTS

Reviewer #1 (Remarks to the Author):

I have nothing else to add, the authors have properly addressed my concerns.

Reviewer #2 (Remarks to the Author):

The authors have addressed my comments with the inclusion of BCKD activity data using an established spectrophotometric assay. I would recommend the acceptance of the manuscript, provided that the following minor deficiency can be corrected during the editorial process, which will require no further review.

The statement on Line 270 of the Results section "BCKD activity levels were consistent with RT-PCR and Western blot results." is imprecise. As shown in Fig. S11, BCKD activity in the liver of *Bckdha*^{-/-} mice treated with 1014 vg/kgEF1 α vector for one month is only 40% of the wild-type activity, which is reduced to 14% after 6 months. Therefore, the degrees of reconstituted hepatic BCKD activity in the *Bckdha*^{-/-} mice are markedly lower than the restoration of the hBCKDHA protein to the wild type levels in the livers of treated *Bckdha*^{-/-} mice (Fig. 3e and Fig.4f). A more accurate description to these effects is needed, which will replace the original statement in the Results section.

The markedly less than robust reconstitution of BCKD activity in AAV8-treated *Bckdha*^{-/-} mice may point to the folding and assembly of the E1 protein as one of the rate-limiting steps in restoring BCKD activity. The recombinantly expressed E1 protein has been shown to depend on molecular chaperones for proper folding and heterotetrameric assembly in cells. Some speculations in the Discussion as to the possible mechanisms for the low BCKD activity in AAV8-treated *Bckdha*^{-/-} mice may strengthen the paper.

RESPONSE TO REVIEWER'S COMMENTS

Reviewer #2 (Remarks to the Author):

The authors have addressed my comments with the inclusion of BCKD activity data using an established spectrophotometric assay. I would recommend the acceptance of the manuscript, provided that the following minor deficiency can be corrected during the editorial process, which will require no further review. The statement on Line 270 of the Results section “BCKD activity levels were consistent with RT-PCR and Western blot results.” is imprecise. As shown in Fig. S11, BCKD activity in the liver of *Bckdha*^{-/-} mice treated with 1014 vg/kgEF1 α vector for one month is only 40% of the wild-type activity, which is reduced to 14% after 6 months. Therefore, the degrees of reconstituted hepatic BCKD activity in the *Bckdha*^{-/-} mice are markedly lower than the restoration of the hBCKDHA protein to the wild type levels in the livers of treated *Bckdha*^{-/-} mice (Fig. 3e and Fig.4f). A more accurate description to these effects is needed, which will replace the original statement in the Results section. The markedly less than robust reconstitution of BCKD activity in AAV8-treated *Bckdha*^{-/-} mice may point to the folding and assembly of the E1 protein as one of the rate-limiting steps in restoring BCKD activity. The recombinantly expressed E1 protein has been shown to depend on molecular chaperones for proper folding and heterotetrameric assembly in cells. Some speculations in the Discussion as to the possible mechanisms for the low BCKD activity in AAV8-treated *Bckdha*^{-/-} mice may strengthen the paper.

Response: We thank the reviewer for this comment concerning the lower reconstitution of BCKD enzyme activity in treated individuals compared to the full restoration of BCKDHA protein. We agree that this is an important finding deserving comment.

We have clearly highlighted this finding at the end of the Results section as follows: “*In these two situations however, the degrees of reconstituted hepatic BCKD activity in the Bckdha*^{-/-} *mice were markedly lower than the restoration of the hBCKDHA protein to the wild type levels in the livers (Fig. 3e and Fig. 4f).*”

Additionally, as requested by the reviewer, some speculations on the possible explanations for this discrepancy were added in the Discussion section with two additional references on the dependence of the E1 protein on molecular chaperones for proper folding as follows: “*The lack of sufficient expression in other tissues, or the production of a non-functional and/or misfolded BCKD enzyme in hepatocytes due to overexpression may also play a role. The recombinantly expressed E1 protein has been shown to depend on molecular chaperones for proper folding and heterotetrameric assembly in cells*^{26,43,44}.”